# The readily-releasable pool dynamically regulates multivesicular release

Jada H Vaden[1], Gokulakrishna Banumurthy[1], Eugeny S Gusarevich[2†],
Linda Overstreet-Wadiche[1]*, Jacques I Wadiche[1]*

[1]Department of Neurobiology, University of Alabama at Birmingham, Birmingham, United States; [2]Department of Fundamental and Applied Physics, Northern (Arctic) Federal University named after M.V. Lomonosov, Arkhangelsk, Russian Federation

**Abstract** The number of neurotransmitter-filled vesicles released into the synaptic cleft with each action potential dictates the reliability of synaptic transmission. Variability of this fundamental property provides diversity of synaptic function across brain regions, but the source of this variability is unclear. The prevailing view is that release of a single (univesicular release, UVR) or multiple vesicles (multivesicular release, MVR) reflects variability in vesicle release probability, a notion that is well-supported by the calcium-dependence of release mode. However, using mouse brain slices, we now demonstrate that the number of vesicles released is regulated by the size of the readily-releasable pool, upstream of vesicle release probability. Our results point to a model wherein protein kinase A and its vesicle-associated target, synapsin, dynamically control release site occupancy to dictate the number of vesicles released without altering release probability. Together these findings define molecular mechanisms that control MVR and functional diversity of synaptic signaling.
DOI: https://doi.org/10.7554/eLife.47434.001

*For correspondence:
lwadiche@uab.edu (LO-W);
jwadiche@uab.edu (JIW)

Present address: †Department of Fundamental and Applied Physics, Northern (Arctic) Federal University named after M.V. Lomonosov, Arkhangelsk, Russian Federation

Competing interests: The authors declare that no competing interests exist.

## Introduction

Excitatory glutamatergic synapses mediate the majority of fast communication between neurons in the brain. While general mechanisms underlying synaptic transmission are well-conserved, there is robust diversity in the synaptic parameters that dictate the reliability and temporal fidelity of transmission between brain regions and species (*Nusser, 2018*). One fundamental parameter that controls synaptic efficacy is the number of vesicles released in response to an action potential. At each presynaptic bouton that contains a single anatomically-defined active zone, it was originally surmised that at most, one neurotransmitter-filled vesicle can be released (*Redman, 1990*). Although the one-site-one-vesicle hypothesis was well accepted for decades, a wealth of evidence now demonstrates that multiple vesicles can be released from a given active zone, a process termed multivesicular release (MVR; *Tong and Jahr, 1994*; *Auger et al., 1998*). MVR is widespread in the rodent CNS and it appears to be the dominant release mode in human cortex (*Rudolph et al., 2015*; *Molnár et al., 2016*). Yet some synapses are limited to the release of a single vesicle per active zone, termed univesicular release (UVR; *Korn et al., 1981*; *Stevens and Wang, 1995*; *Silver, 2003*; *Biró et al., 2005*) and other synapses switch release modes in an activity-dependent manner (*Bender et al., 2009*; *Higley et al., 2009*; *Pulido et al., 2015*). Understanding the mechanisms that determine whether an active zone supports UVR or MVR is important to understand synaptic diversity as well as to develop unifying models of presynaptic function that account for multiple presynaptic parameters.

The number of vesicles released from an active zone is the product of vesicle release probability (Pr) and the number of vesicles available for release. While MVR typically correlates with Pr, experimental evidence supporting Pr as the sole determinant of MVR is mixed. At many synapses, including the climbing fiber (CF) to Purkinje cell (PC) synapse in the cerebellum, manipulating Pr alters the

**eLife digest** Our nervous system allows us to rapidly sense and respond to the world around us via cells called neurons that relay electrical signals around the brain and body. When an electrical impulse travelling along one neuron reaches a junction – called a synapse – with a neighboring neuron, it stimulates small containers known as vesicles from the first cell to release their contents into the synapse. These contents then travel across to the neighboring cell and may generate a new electrical impulse.

The number of vesicles at a synapse that are ready to be released varies from one to ten. The more vesicles the neuron releases, the more likely the second cell will produce an electrical signal of its own. However, not all electrical signals reaching a synapse stimulate vesicles to be released and some signals only release a single vesicle.

What determines how many vesicles are released by a single electrical signal? Some vesicles have a higher likelihood of being released than others, but this "eagerness" does not always predict how many vesicles an individual synapse will actually discharge. Now, Vaden et al. have used brain tissue from mice to test an alternative possibility: the simple idea that the number of vesicles available at the synapse affects how many vesicles are released without altering their eagerness for release.

Vaden et al. found that activating an enzyme called protein kinase A increased the number of vesicles released from synapses without changing how likely individual vesicles were to be released. Inhibiting protein kinase A also did not change individual vesicle's eagerness to be released, but did decrease the number of vesicles that were discharged. Further experiments found that protein kinase A modifies a molecule on the surface of vesicles, known as synapsin, which controls the number of vesicles that are available for release.

These findings show that the number of vesicles released at a synapse is controlled by two independently regulated parameters: the number of vesicles that are available, as well as how eager individual vesicles are to be released. The ability of neurons to communicate with each other is disrupted in autism spectrum disorders, Alzheimer's disease and many other diseases. Learning how neurons communicate in healthy brains will help us understand what happens in the neurons of individuals with these conditions.

DOI: https://doi.org/10.7554/eLife.47434.002

degree of MVR in the expected manner (*Tong and Jahr, 1994*; *Wadiche and Jahr, 2001*; *Bender et al., 2009*; *Higley et al., 2009*; *Nahir and Jahr, 2013*). However, some high Pr (0.8–0.9) synapses display exclusively UVR (*Silver, 2003*; *Murphy et al., 2004*), whereas some low Pr (0.13–0.33) synapses can exhibit MVR (*Oertner et al., 2002*; *Taschenberger et al., 2002*; *Foster et al., 2005*; *Christie and Jahr, 2006*). This illustrates that Pr alone is insufficient to dictate vesicle release mode, pointing to the potential contribution of vesicle availability. What determines vesicle availability? Anatomical analysis shows that single active zones contain a variable number of docked vesicles that may represent release-competent vesicles (*Schikorski and Stevens, 1997*). Yet it is debated whether anatomically-defined docked vesicles correspond to the population of vesicles that comprise the readily-releasable pool (RRP), a functional measure that defines the available vesicles across all active zones (*Dobrunz and Stevens, 1997*; *Schikorski and Stevens, 2001*; *Rizzoli and Betz, 2004*; *Moulder and Mennerick, 2005*). An emerging idea is that the number of docking sites (i.e. release sites) within an active zone is constant but their occupancy by release-competent vesicles is dynamic (*Pulido and Marty, 2017*).

Surprisingly, whether the RRP dictates the number of vesicles released per active zone has not been addressed. As the key parameter controlling the number of available vesicles, we predict that the RRP is a critical determinant release mode. To overcome the constraints and assumptions of anatomical measures, we indirectly measured the synaptic cleft glutamate concentration as a proxy for the number of vesicles released per active zone while separately measuring Pr and the RRP. We find that cyclic AMP-dependent (cAMP) protein kinase A (PKA), a canonical pathway that induces various forms of presynaptic plasticity and increases MVR (*Huang et al., 1994*; *Salin et al., 1996*; *Chen and Regehr, 1997*; *Moulder et al., 2008*; *Bender et al., 2009*; *Midorikawa and Sakaba, 2017*), bidirectionally modifies release mode via regulation of the RRP. We show that PKA regulation of MVR can

be dissociated from its effects on Pr, and requires synapsins, a family of proteins that are well-known to control the RRP. Our results support a model wherein the release mode is controlled by release site occupancy upstream of Pr, potentially explaining the paradoxical reports of MVR in conditions of high and low Pr. By dissociating the requirement of high Pr from MVR, these results provide new insight into molecular mechanisms that control MVR and illustrate how diverse modes of synaptic transmission can arise from common signaling pathways.

## Results

### cAMP/PKA stimulation shifts the balance of vesicle release from UVR to MVR mode without affecting Pr

We studied regulation of neurotransmitter release at the climbing fiber (CF) to Purkinje cell (PC) synapse using 0.5 mM extracellular $Ca^{2+}$ that constrains transmission to the release of zero or one vesicle with each action potential (UVR). As in other synapses, the adenylyl cyclase activator forskolin (fsk; *Figure 1A* left) potentiated neurotransmitter release as measured by ~33% increase in the peak amplitude of EPSCs (*Salin et al., 1996*; *Chen and Regehr, 1997*; *Sakaba and Neher, 2001*; *Bender et al., 2009*; *Cousin and Evans, 2011*; *Midorikawa and Sakaba, 2017*). However, the fsk-mediated increase of EPSCs was not accompanied by a change in the paired-pulse ratio (PPR; *Figure 1A* middle and right panels). Reduced PPR is often used as a proxy for the increase in Pr that accompanies the cAMP/PKA-mediated enhancement of coupling between $Ca^{2+}$ channels and synaptic vesicles (*Salin et al., 1996*; *Chen and Regehr, 1997*; *Chevaleyre et al., 2007*; *Pelkey et al., 2008*; *Ariel et al., 2012*; *Midorikawa and Sakaba, 2017*) because a change in Pr during the first stimulus causes predictable changes in the second response (*Dobrunz and Stevens, 1997*; *Zucker and Regehr, 2002*). Similarly, we found that the PKA activator 6-Bnz-cAMP (6-Bnz; *Figure 1B* left) increased CF-evoked EPSCs without altering PPR (*Figure 1B* middle and right panels). Consistent with a lack of change in Pr, the coefficient of variation (CV) of EPSCs was not altered by either fsk or 6-Bnz treatment (104.1 ± 44.9% and 101.9 ± 40.1%; n = 5 and 10, p=0.93 and 0.96, one sample t-test), a similar result to reducing the amplitude of AMPA receptor (AMPAR)-mediated EPSCs with NBQX (97.6 ± 38.3%; n = 4, p=0.95). In contrast, altering Pr by manipulating extracellular $Ca^{2+}$, robustly changed the CV (216.4 ± 27.6% and 49.2 ± 8.6%, respectively; p=0.01 and 0.03, n = 5 and 3; *Figure 1—figure supplement 1*).

Finding no evidence for changes in Pr, we asked whether the enhancement of CF EPSC amplitudes occurs postsynaptically through regulation of AMPA receptors (*Kerchner and Nicoll, 2008*). However, 6-Bnz caused a similar increase in EPSC amplitude with the protein kinase A inhibitor peptide (Pki) in the recording pipette (*Figure 1B* right panel, filled symbols), making it unlikely that PKA simply increased AMPAR number or conductance. Therefore, we considered two alternative processes to account for our results. Activation of PKA could increase the number of active zones, reminiscent of mechanisms underlying presynaptic unsilencing/silencing (*Moulder et al., 2004*; *Cousin and Evans, 2011*). Alternatively, PKA could increase the number of vesicles released per active zone, shifting synapses towards MVR and increasing the synaptic glutamate concentration (*Tong and Jahr, 1994*; *Wadiche and Jahr, 2001*; *Oertner et al., 2002*; *Singer et al., 2004*; *Bender et al., 2009*; *Higley et al., 2009*; *Rudolph et al., 2015*). We distinguished between these possibilities by comparing EPSC inhibition by a 'low-affinity antagonist' (LAA) before and after manipulation of cAMP/PKA signaling. Only changes that alter the synaptic glutamate concentration will alter the magnitude of LAA inhibition, whereas changes in the number of active zones or the number of receptors will not affect inhibition by LAAs (*Clements, 1996*).

First, we confirmed the assumption that varying the number of functional release sites does not affect the LAA block. This is an important control due to the potential for glutamate to pool between neighboring sites altering the glutamate concentration time course and thus the degree of LAA inhibition. We used increasing stimulation frequency to reduce the number of functional release sites by vesicle depletion. In UVR conditions (0.5 mM $Ca^{2+}$), we used variance/mean analysis to confirm that the reduction in EPSC amplitude is a result of a decrease in the number of active release sites (*Figure 1—figure supplement 2A*) (*Foster and Regehr, 2004*). The LAA, kynurenic acid (KYN; 1 mM), inhibited EPSCs equally despite a ~ 30% reduction in amplitude with increasing stimulation frequencies (*Figure 1—figure supplement 2B and C*; see also *Rudolph et al., 2011*). As expected,

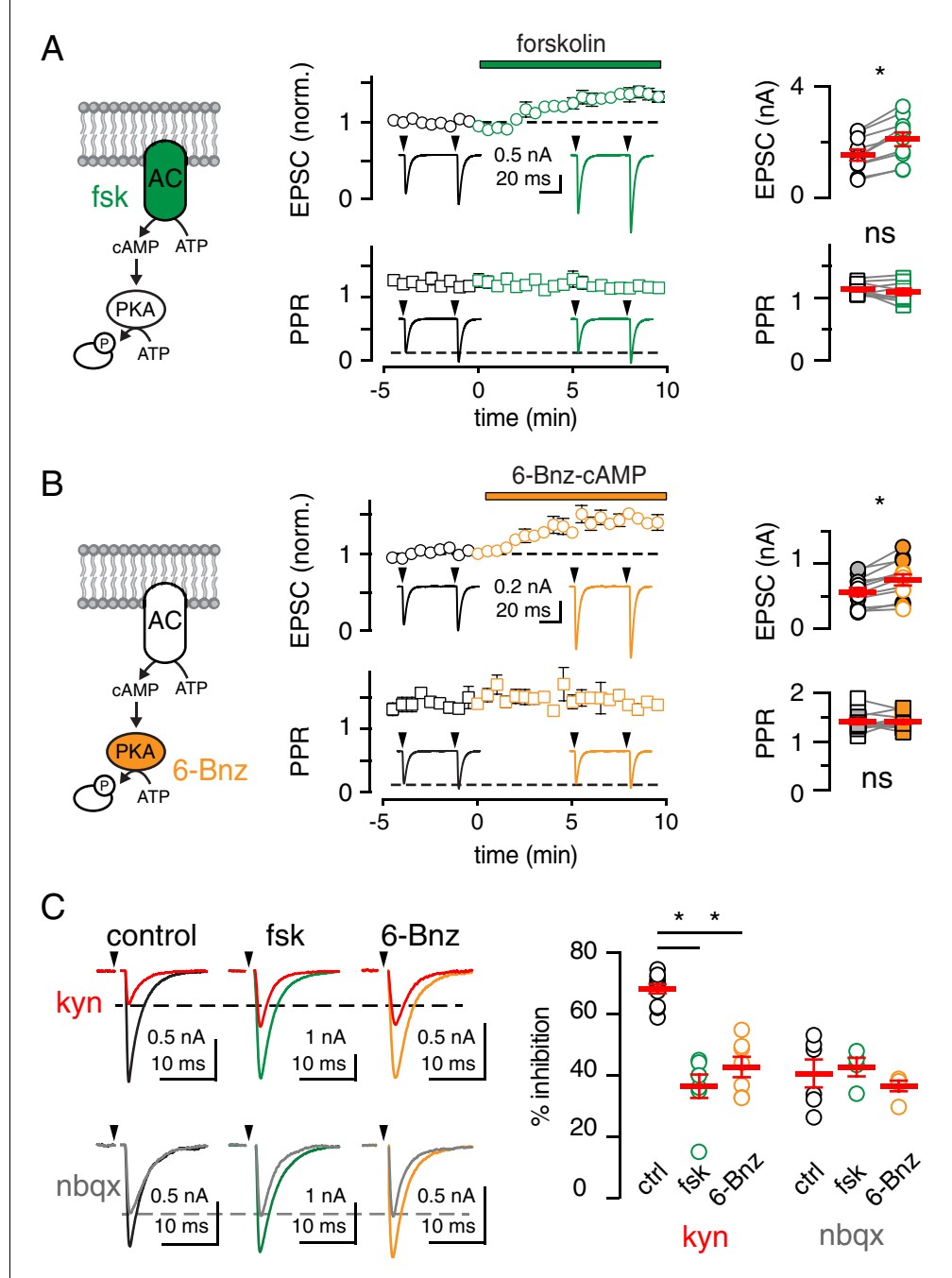

**Figure 1.** cAMP/PKA activation shifts vesicle release mode from UVR to MVR. (**A and B**, left). Fsk (50 μM) was used to stimulate cAMP production by adenylyl cyclase (AC, green) and the cAMP analog 6-Bnz (20 μM) was used to activate protein kinase A (PKA, orange). (**A and B**, middle) Time course of CF-PC EPSC amplitude (top: normalized, circles) and paired pulse ratio (bottom: PPR with an inter-stimulus interval = 50 ms, squares) following bath application of fsk (green) or 6-Bnz (orange), respectively. Insets: representative and normalized traces show the increase in amplitude and lack of change in PPR. (**A and B**, right) Fsk increased the EPSC amplitude (from 1.6 ± 0.2 nA to 2.1 ± 0.3 nA; p<0.0001, paired t-test) without changing the PPR (from 1.1 ± 0.03 to 1.1 ± 0.05, n = 10 each; p=0.27). Likewise, 6-Bnz increased the EPSC amplitude (from 0.6 ± 0.1 nA to 0.7 ± 0.1 nA; p=0.0009, paired t-test) without changing the PPR (from 1.4 ± 0.07 to 1.4 ± 0.05, n = 10 each; p=0.68). Filled symbols indicate inclusion of the PKA inhibitory peptide PKi (1 μM) in the patch pipette. For this data set: EPSC amplitude (from 0.6 ± 0.1 nA to 0.8 ± 0.1 nA; p=0.02 and PPR from 1.4 ± 0.03 to 1.4 ± 0.07, n = 6 each; p=0.51, paired t-test). (**C**, left) Superimposed EPSCs (0.5 mM Ca²⁺/10 mM Mg²⁺) before and after bath application of 250 μM KYN (top) or

*Figure 1 continued on next page*

*Figure 1 continued*

100 nM NBQX (bottom) in the absence or presence of fsk or 6-Bnz. (C, right) Both fsk and 6-Bnz reduced % EPSC inhibition by kyn from 68.3 ± 1.5% in control (n = 11) to 36.5 ± 3.8% in fsk (n = 7; p<0.0001) and to 42.8 ± 3.3% in 6-Bnz (n = 7; p<0.0001; ANOVAs followed by Holm-Sidak post-tests). In contrast, neither fsk nor 6-Bnz affected % EPSC inhibition by NBQX (control: 40.7 ± 4.6%, n = 6; fsk: 42.7 ± 3.0%, n = 4, p=0.98; 6-Bnz: 36.6 ± 1.7%, n = 5; p=0.92, ANOVA). Asterisks denote statistical significance.

DOI: https://doi.org/10.7554/eLife.47434.003

The following figure supplements are available for figure 1:

**Figure supplement 1.** CF-PC EPSCs CV analysis.

DOI: https://doi.org/10.7554/eLife.47434.004

**Figure supplement 2.** Reducing release site number does not alter KYN block.

DOI: https://doi.org/10.7554/eLife.47434.005

the high-affinity antagonist NBQX (100 nM) that is insensitive to cleft glutamate concentration also blocked EPSCs similarly (*Figure 1—figure supplement 2C*). Thus, varying the number of active release sites does not influence the synaptic glutamate concentration.

We next used LAA-inhibition of EPSCs to determine whether fsk or 6-Bnz potentiation results from a change in the number of vesicles released per synaptic site or in the number of active sites. We found that both activators markedly reduced KYN inhibition without affecting NBQX inhibition, strongly suggesting that the average synaptic glutamate concentration is higher following stimulation of cAMP/PKA signaling (*Figure 1C*). These results are not likely due to increased glutamate spillover from nearby active sites because 1) recordings were performed in low extracellular $[Ca^{2+}]$ that supports sparse synaptic activation and 2) the kinetics of the potentiated EPSCs were unchanged (EPSC decay control: 3.9 ± 0.3 ms; fsk : 4.1 ± 0.4 ms; 6-Bnz: 3.9 ± 0.3 ms; p=0.5). Together these data show that stimulation of the cAMP/PKA pathway increases the synaptic glutamate concentration without altering Pr.

## cAMP/PKA activation increases the initial RRP but does not change quantal size

A larger glutamate transient could be a consequence of a higher glutamate concentration within each vesicle. To test this idea, we recorded, quantal-like, $Sr^{2+}$-evoked asynchronous EPSCs (aEPSCs) following PKA activation. 6-Bnz increased the frequency of aEPSCs (from 1.3 ± 0.2 to 3.5 ± 0.8 Hz; n = 6, p=0.04), an effect that was evident in the individual sweeps and in the cumulative probability histogram (*Figure 2A and B*). 6-Bnz had no effect on the cumulative amplitude probability histogram, aEPSC average amplitude (55 ± 4 pA and 56 ± 3 pA; n = 6, p=0.7) or aEPSC kinetics (rise: 0.21 ± 0.02 ms and 0.22 ± 0.02 ms; decay: 1.8 ms ± 0.2 and 1.8 ms ± 0.2; n = 6, p=0.5; *Figure 2C*) demonstrating that PKA stimulation does not affect quantal size but, rather, may increase the total number of vesicles available for release. An increase in the frequency of aEPSCs could thus indicate an increase in the size of the initial RRP.

To test whether cAMP/PKA stimulation increases the initial RRP, we quantified the cumulative EPSC during a long train of stimuli. This analysis assumes that depression of the EPSC during the train occurs as the RRP is depleted. The product of the total number of releasable vesicles and the quantal size can be estimated from the y-intercept of the linear regression fit to the linear portion of the cumulative EPSC amplitude plot (*Schneggenburger et al., 1999*). In the presence of KYN (3 mM) to relieve AMPAR saturation, train stimulation initially facilitated CF-EPSCs followed by gradual depression (100 Hz; *Figure 2D*). Both the amplitude and the cumulative EPSC were potentiated by 6-Bnz, but neither the initial facilitation nor the gradual depression was altered (*Figure 2—figure supplement 1*). Linear regression of the last EPSCs (see Materials and methods) showed that PKA activation enhanced the size of the RRP by ~30% (5.0 ± 0.9 nA and 6.7 ± 1.2 nA, n = 6, p=0.04) without changing Pr (calculated as the ratio of the initial EPSC to the y-intercept of the cumulative EPSC; 0.03 ± 0.005 and 0.03 ± 0.003, p=0.8; *Figure 2E*). These results allow comparisons between conditions rather than absolute numbers, noting that the measures described here are equivalent to the effective or functional releasable pool (the portion of vesicles accessed by action potentials) and this effective RRP is likely comprised of multiple subpools with variable rates of release (*Thanawala and Regehr, 2013*). Nevertheless, these data show that stimulation of the cAMP/PKA pathway under

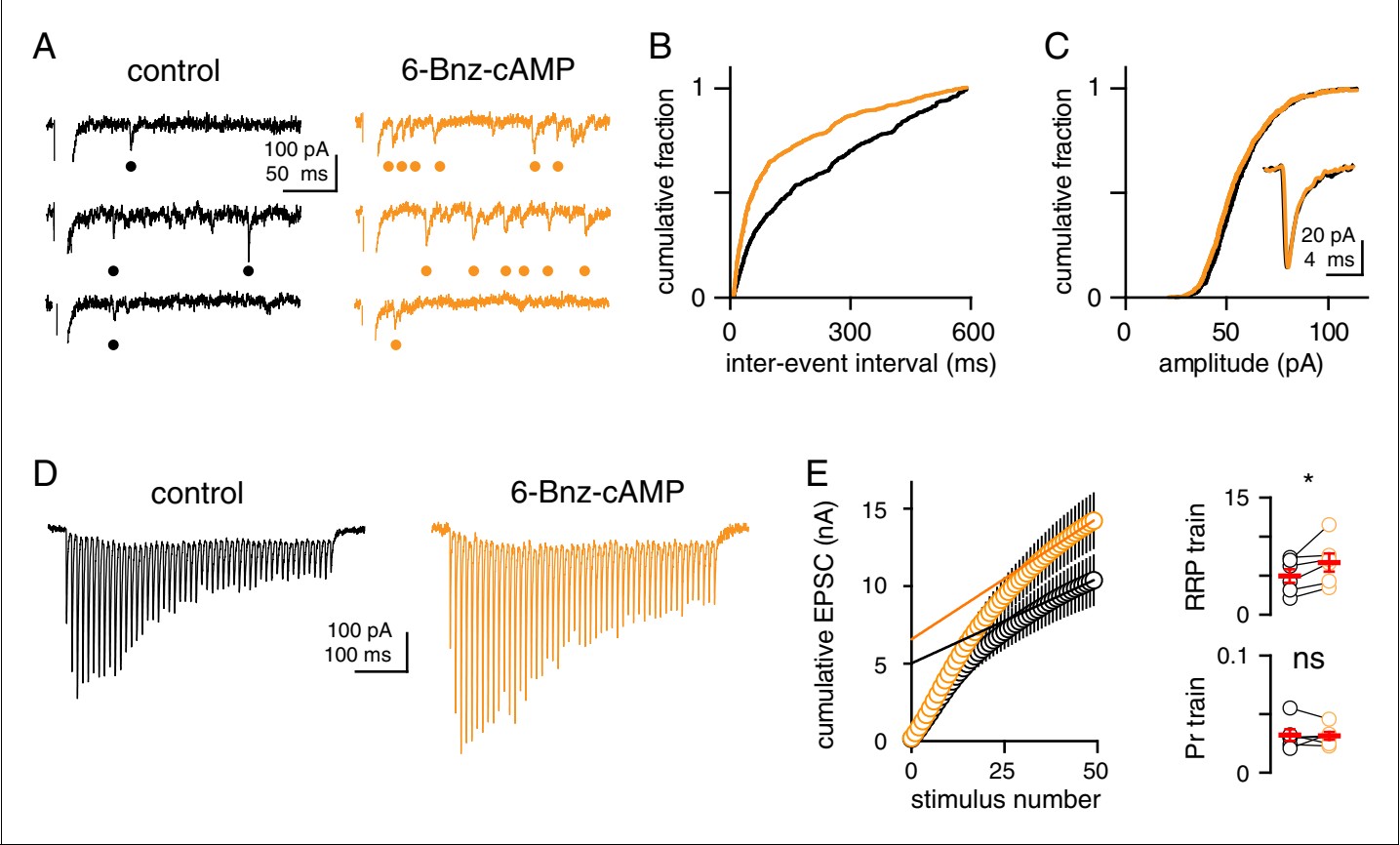

**Figure 2.** cAMP/PKA activation does not change quantal size but increases the RRP. (**A**) Representative sweeps of $Sr^{2+}$-evoked asynchronous EPSCs (aEPSCs) before (black) and after (orange) application of 6-Bnz (20 µM). Bullets denote detected events. (**B**) Distribution of aEPSC inter-event intervals before and after bath application of 6-Bnz (n = 6; p<0.0001, KS test). (**C**) Distribution of aEPSC amplitudes in control conditions (black) and 6-Bnz (20 µM, orange), compared with the (n = 6; p>0.99, KS test). Inset shows average aEPSCs. (**D**) Representative EPSCs recorded in response to CF stimulation at 100 Hz for 500 ms before (black) and after (orange) 6-Bnz treatment. (**E**, left) Cumulative EPSC amplitude plotted as a function of stimulus number before (black) and after (orange) 6-Bnz treatment. A line was fit to the final 5 EPSCs in each condition and the y-intercept of this line was used to estimate the RRP. (**E**, right).

DOI: https://doi.org/10.7554/eLife.47434.006

The following figure supplement is available for figure 2:

**Figure supplement 1.** cAMP activation does not affect short-term plasticity.
DOI: https://doi.org/10.7554/eLife.47434.007

UVR conditions (0.5 mM $Ca^{2+}$) increased the synaptic glutamate concentration by increasing the size of the RRP rather than affecting Pr or quantal size.

## cAMP/PKA activation is occluded when MVR is prevalent

Since manipulations that affect pool size may also influence Pr, it is necessary to assay putative changes of pool size across various Pr conditions (*Neher, 2015*). Furthermore, the propensity for MVR at individual synapses typically correlates with Pr (*Tong and Jahr, 1994*; *Wadiche and Jahr, 2001*; *Oertner et al., 2002*; *Christie and Jahr, 2006*; *Bender et al., 2009*), thus we tested the effects of cAMP/PKA stimulation under conditions of high Pr when MVR predominates (*Silver et al., 1998*; *Wadiche and Jahr, 2001*; *Foster and Regehr, 2004*; *Rudolph et al., 2011*). In the presence of KYN to relieve AMPAR saturation, increasing extracellular $Ca^{2+}$ (to 2.5 mM) results in EPSCs that show marked paired-pulse depression to a pair of stimuli (50 ms interval), consistent with high Pr (*Wadiche and Jahr, 2001*; *Harrison and Jahr, 2003*; *Wong et al., 2003*; *Foster and Regehr, 2004*; *Foster et al., 2005*). Surprisingly, neither fsk nor 6-Bnz application altered EPSC amplitude (99 ± 2% and 100 ± 1% of control with fsk and 6-Bnz, respectively) or short-term plasticity (PPR = 0.15 ± 0.01,

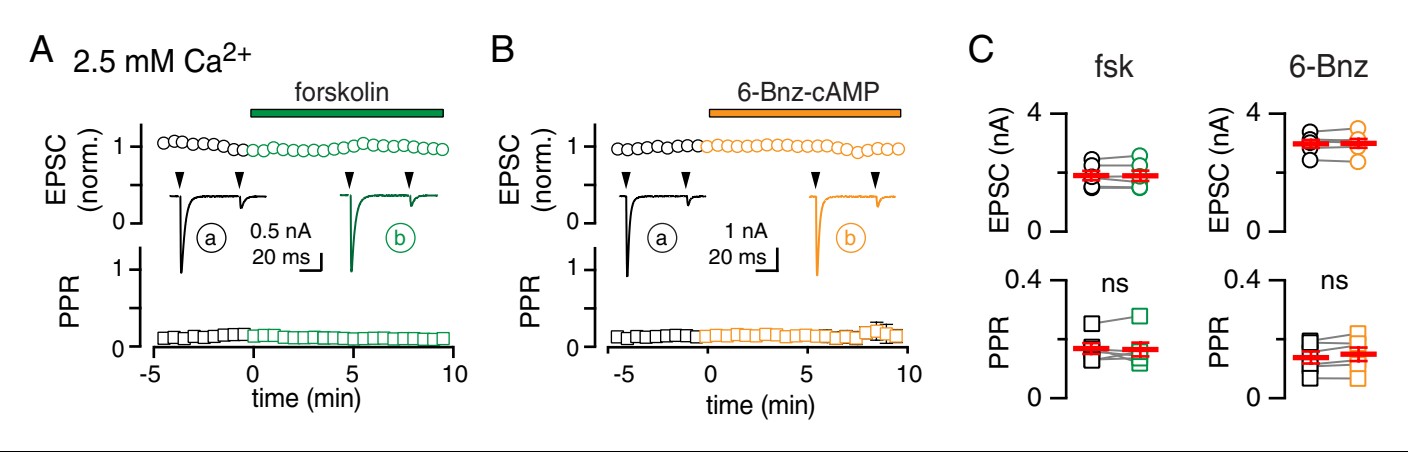

**Figure 3.** cAMP/PKA activation is occluded when MVR is prevalent. (**A and B**) Time course of CF-PC EPSC amplitude (top: normalized, open circles) and PPR (bottom: open squares) during bath application of fsk (green) or 6-Bnz (orange). Insets: representative traces showing the lack of change in amplitude or PPR. Recordings performed in 2.5 mM $Ca^{2+}$ and 1.3 mM $Mg^{2+}$. (**C**) Fsk had no effect on EPSC amplitude ($1.9 \pm 0.2$ nA in control versus $1.9 \pm 0.2$ nA in fsk; n = 6, p=0.87, paired t-test) or the PPR ($0.17 \pm 0.02$ in control versus $0.17 \pm 0.02$ in fsk, n = 6, p=0.82, paired t-test). Likewise, 6-Bnz had no effect on EPSC amplitude ($3.0 \pm 0.2$ nA in control versus $3.0 \pm 0.1$ nA in 6-Bnz; n = 6, p=0.64, paired t-test) or PPR ($0.14 \pm 0.02$ in control versus $0.15 \pm 0.03$ n=6; p=0.08, paired t-test).

DOI: https://doi.org/10.7554/eLife.47434.008

$0.17 \pm 0.02$ and $0.15 \pm 0.03$ in control, fsk and 6-Bnz, respectively, n = 12, 6, and 6; *Figure 3*). Thus, cAMP/PKA signaling does not further potentiate existing MVR, suggesting that the effects of cAMP/PKA on vesicular release may be occluded when MVR is prevalent.

## PKA-inhibition shifts vesicle release mode from MVR to UVR

Extracellular $Ca^{2+}$ not only regulates Pr, but can also alter the size of the RRP (*Schneggenburger et al., 1999*; *Thanawala and Regehr, 2013*). We hypothesized that in high Pr conditions (2.5 mM $Ca^{2+}$) MVR is maximized by both an increase in Pr and an increase the size of the RRP, thus occluding further effects of PKA activation. We tested this idea by preincubating slices in either KT5720 (1 µM), an active-site directed inhibitor that occupies PKA's ATP binding pocket and prevents phosphorylation of its substrates, or Rp-8Br-cAMPs (50 µM), a cAMP analog that prevents the dissociation of PKA's regulatory subunits from its catalytic subunits. Both compounds are expected to lower the effective PKA concentration at terminals, and indeed, both reduced MVR. We found that CF-PC EPSC inhibition by KYN (1 mM) in control ($26 \pm 4\%$, n = 10) was increased in slices pretreated with KT5720- or 8-Br-cAMPs ($47 \pm 3\%$ and $42 \pm 2\%$, n = 6 and 7, p=0.0004 and p=0.0048, respectively; *Figure 4A and B*). In contrast, NBQX (100 nM) inhibition was similar with all treatments (control: $34 \pm 3\%$, KT5720: $33 \pm 2\%$, 8-Br-cAMPs: $33 \pm 5\%$; n = 9, 4, and 3, p=0.97; *Figure 4A and B*). Consistent with our results shown above, the reduction in MVR did not result from reduced Pr, which we measured with two methods. The magnitude of paired-pulse depression (PPD) will approximate Pr at very brief interstimulus intervals if the second stimulus occurs when there has not been sufficient time for recovery (*Dittman and Regehr, 1998*). Since little recovery of the EPSC occurred at 10 ms ($7.6 \pm 0.5\%$; *Figure 4—figure supplement 1A*), we estimated Pr using the assumptions of a simple depletion model and found it to be unchanged with PKA inhibition (*Figure 4—figure supplement 1B*). Similarly, PPR was unchanged across a range of interstimulus intervals (*Figure 4C*). Together, these data suggest that PKA-mediated changes in MVR are independent from Pr, since neither KT5720 nor 8-Br-cAMPs altered these widely-used estimates of Pr.

To fully explore the limits of this regulation, we systematically varied the number of vesicles released per synapse by changing extracellular $[Ca^{2+}]$ and measured how blocking PKA activity affects KYN inhibition of EPSCs. As shown in *Figure 4D*, KYN inhibition was well described by a curve with a minimum as extracellular $Ca^{2+}$ was increased, presumably reflecting maximal MVR and elevated synaptic glutamate concentration. Curve fits of the average data exhibited similar IC50s and slopes in control and KT5720-treated slices, indicating that inhibition of PKA signaling did not

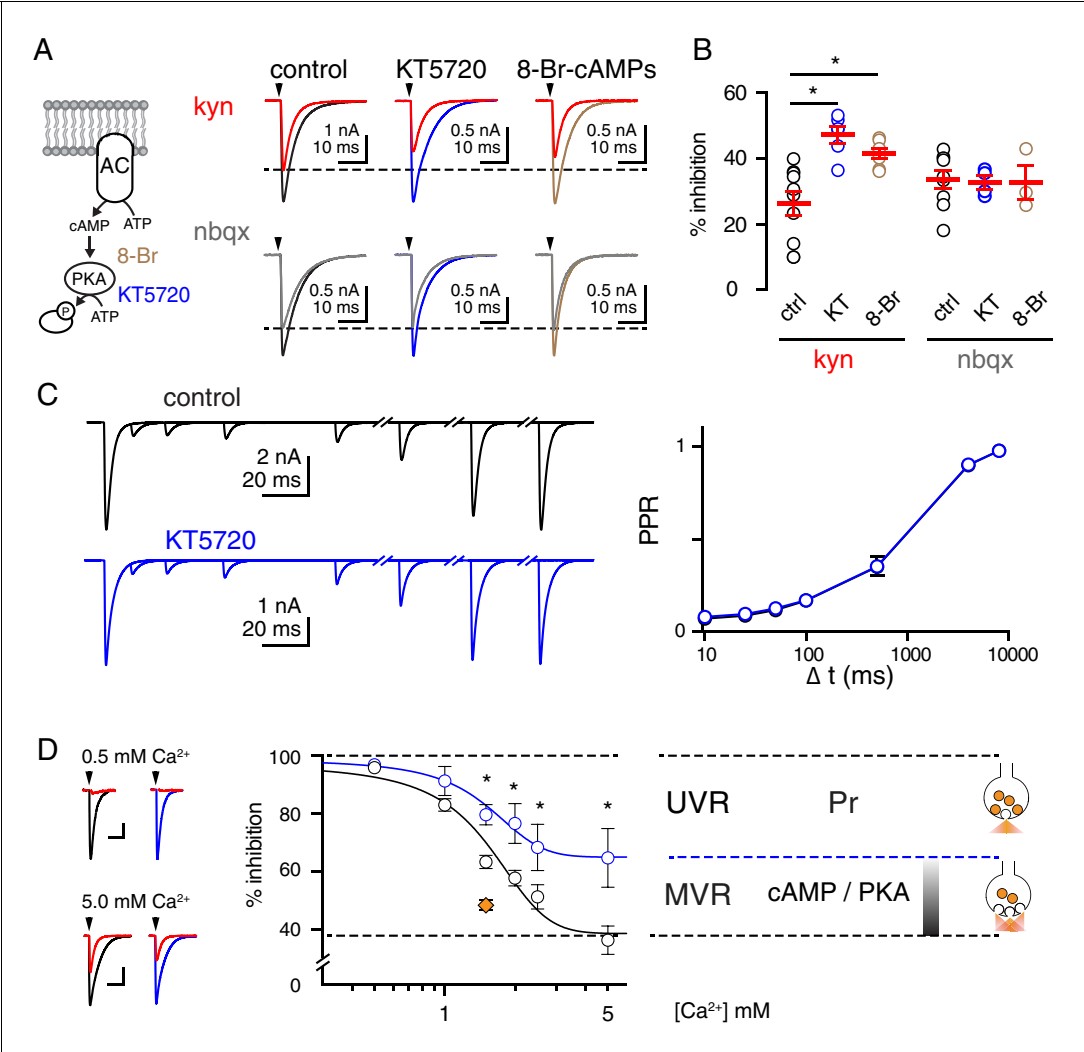

**Figure 4.** PKA-inhibition shifts vesicle release mode from MVR to UVR. (**A**, left) The inactive cAMP analog 8-Br-cAMPs (8-Br, brown) was used to prevent activation of PKA and KT5720 (blue), a small molecule that occupies PKA's ATP-binding site, was used to inhibit phosphorylation of substrates by PKA. (**A**, right) Superimposed EPSCs before and after bath application of kynurenic acid (1 mM kyn, top) or NBQX (100 nM, bottom) in control slices and those incubated with KT5720 (1 μM) or 8-Br-cAMPs (50 μM) for 90–120 min. Experiments were performed in 2.5 mM $Ca^{2+}$ and 1.3 mM $Mg^{2+}$. Membrane potential has held between −10 and −20 mV to reduce the amplitude of CF-EPSCs. (**B**) Incubation with KT5720 or 8-Br-cAMPs increased the %inhibition by kyn (control: 26 ± 4%, KT5720: 47 ± 3%, 8-Br-cAMPs: 42 ± 2%; n = 10, 6, and 7, p=0.0004 and p=0.0048, ANOVA and Holm-Sidak post-tests). Incubation with KT5720 or 8-Br-cAMPs had no effect on % inhibition by NBQX (control: 34 ± 3%, KT5720: 33 ± 2%, 8-Br-cAMPs: 33 ± 5%; n = 9, 4, and 4; p=0.97, ANOVA). Asterisks denote statistical significance. (**C**, left) Representative traces in control (black) and following KT5720 incubation (blue). There was no change in the PPR (**C**, right) at any interstimulus intervals (Δt). PPR in control 0.07 ± 0.009, 0.09 ± 0.01, 0.12 ± 0.01, 0.17 ± 0.02, 0.35 ± 0.03, 0.90 ± 0.01, 0.98 ± 0.01 and in KT5720 0.08 ± 0.005, 0.1 ± 0.01, 0.13 ± 0.01, 0.17 ± 0.02, 0.35 ± 0.05, 0.90 ± 0.01, 0.98 ± 0.01 for interstimulus intervals of 10, 25, 100, 500, 4000, 8000 ms, respectively; n = 8 for each; p=0.8, repeated measures ANOVA. (**D**, left) Superimposed EPSCs before and after bath application of KYN (3 mM, red) in the indicated extracellular $Ca^{2+}$ in control slices (black) or those incubated with KT5720 (blue). Scale bars: 10 pA, 10 ms; and 1 nA, 10 ms. (**D**, right) Semi-log plot of the inhibition by KYN (3 mM) as a function of extracellular $Ca^{2+}$ in control (black) and KT5720-treated slices (blue). The $Ca^{2+}$-insensitive inhibition by KYN determined from the best fit to a four-parameter dose response curve was 38.5% in control and 64.9% in KT5720-treated slices suggesting that cAMP/PKA signaling controls 43% of the total KYN inhibition. The orange diamond (1.5 mM $Ca^{2+}$) denotes inhibition by 3 mM KYN in the presence of the PKA activator 6-Bnz.

DOI: https://doi.org/10.7554/eLife.47434.009

The following figure supplements are available for figure 4:

**Figure supplement 1.** No change in PPR using a simple depletion model.

DOI: https://doi.org/10.7554/eLife.47434.010

**Figure supplement 2.** PKA inhibition and PPR changes with extracellular $Ca^{2+}$.

DOI: https://doi.org/10.7554/eLife.47434.011

*Figure 4 continued on next page*

*Figure 4 continued*

**Figure supplement 3.** Inhibition of cAMP/PKA does not affect quantal content.

DOI: https://doi.org/10.7554/eLife.47434.012

alter the $Ca^{2+}$-dependence of Pr. Indeed, while the PPR (50 ms interstimulus interval, ISI) was highly sensitive to changes in extracellular $Ca^{2+}$ reflecting changes in Pr, it was not affected by KT5720 (*Figure 4—figure supplement 2*). KT5720 did cause a 43% increase in KYN's ability to inhibit EPSCs at the highest $Ca^{2+}$ concentration tested (64.7 ± 5.1%, n = 4 and 36.2 ± 4.9%, n = 5; in KT5720 and control, respectively; *Figure 4D*). This indicates that raising extracellular $Ca^{2+}$ causes a PKA-mediated increase in the number of vesicles released that is *independent* of the expected $Ca^{2+}$-mediated change in Pr.

These results suggest that MVR is controlled by two $Ca^{2+}$-dependent processes: 57% of MVR results from changes in Pr whereas 43% of MVR is under the direct control of PKA activity (*Figure 4D* right). As an additional control, we found that inhibition of PKA with KT5720 had no effect on the size or kinetics of $Sr^{2+}$ evoked aEPSCs but reduced their frequency (*Figure 4—figure supplement 3*). Together, bi-directional regulation of the glutamate concentration transient (*Figure 1* and *Figure 4*) and of the aEPSC frequency (*Figure 2* and *Figure 4—figure supplement 3*) suggests that PKA signaling modulates MVR by controlling the number of vesicles available for release that may be part of the RRP.

## cAMP/PKA inhibition reduces the size of the RRP without affecting Pr

To test whether inhibition of PKA alters the size of the RRP in 2.5 mM $Ca^{2+}$, we used two methods with different assumptions to analyze CF-EPSCs during train stimulation (*Thanawala and Regehr, 2013*). In the presence of KYN (3 mM) to relieve AMPAR saturation, the average amplitude of the first EPSCs in KT5720-treated slices was half of those in control and the trains scaled proportionately (12.3 ± 0.7 nA and 6.0 ± 0.8 nA, n = 7 and 5, unpaired t-test, p=0.0002; *Figure 5A and B*). We first plotted the cumulative EPSC amplitude versus stimulus number and used the y-intercept of a fit to the final stimuli in the train to estimate the RRP, as in *Figure 2*. This analysis assumes constant vesicle replenishment throughout train (*Schneggenburger et al., 1999*). This estimate of the RRP (RRP train) showed that PKA inhibition reduced the RRP to approximately half of that in control (*Figure 5C*). We also used a second method that assumes vesicle replenishment is negligible early in the train when EPSC amplitudes decay linearly (*Figure 5B*, inset; *Elmqvist and Quastel, 1965*; *Taschenberger et al., 2002*). This approach is well-justified in 2.5 mM $Ca^{2+}$ because CF depression results from depletion of the RRP (*Silver et al., 1998*; *Foster and Regehr, 2004*) and we found that replenishment after the train is negligible at short intervals *Figure 4—figure supplement 1*. A plot of the EPSC amplitudes versus the cumulative EPSC showed that the first response almost completely depletes the RRP and the x-intercept of a line through the first two points corresponds to a second estimate of the RRP (RRP EQ). This method also showed that the RRP in KT5720-treated slices was approximately half of that in control (*Figure 5D*). In both analyses, the ratio of the initial EPSC to the cumulative EPSC plot intercept provides an estimate of Pr (*Figure 5C and D*). Consistent with estimates of Pr derived from PPD, neither $Pr_{train}$ nor $Pr_{EQ}$ differed significantly between control and KT5720-treated slices.

In summary, we used four separate assays to rule out the possibility that PKA activity alters the number of vesicles released via changes in Pr. CV analysis, the ratio of cumulative EPSC/EPSC1 using two methods and the PPR model of depletion all support the conclusion that Pr is unaffected. Conversely, both methods of estimating the number of vesicles available for AP-mediated release and bi-directional regulation of aEPSCs point to a mechanism involving PKA-mediated changes in the RRP. Altogether, these data suggest that PKA activity alters the number of vesicles released per active zone independent of the likelihood that a release-competent vesicle fuses with the presynaptic membrane. In other words, PKA activity changes the RRP without altering Pr.

## PKA-mediated regulation of MVR requires synapsin

Among the many presynaptic proteins that can be directly or indirectly regulated by PKA (*Sudhof, 2004*), the A-domain of synapsin is an excellent PKA substrate that is well-known to regulate

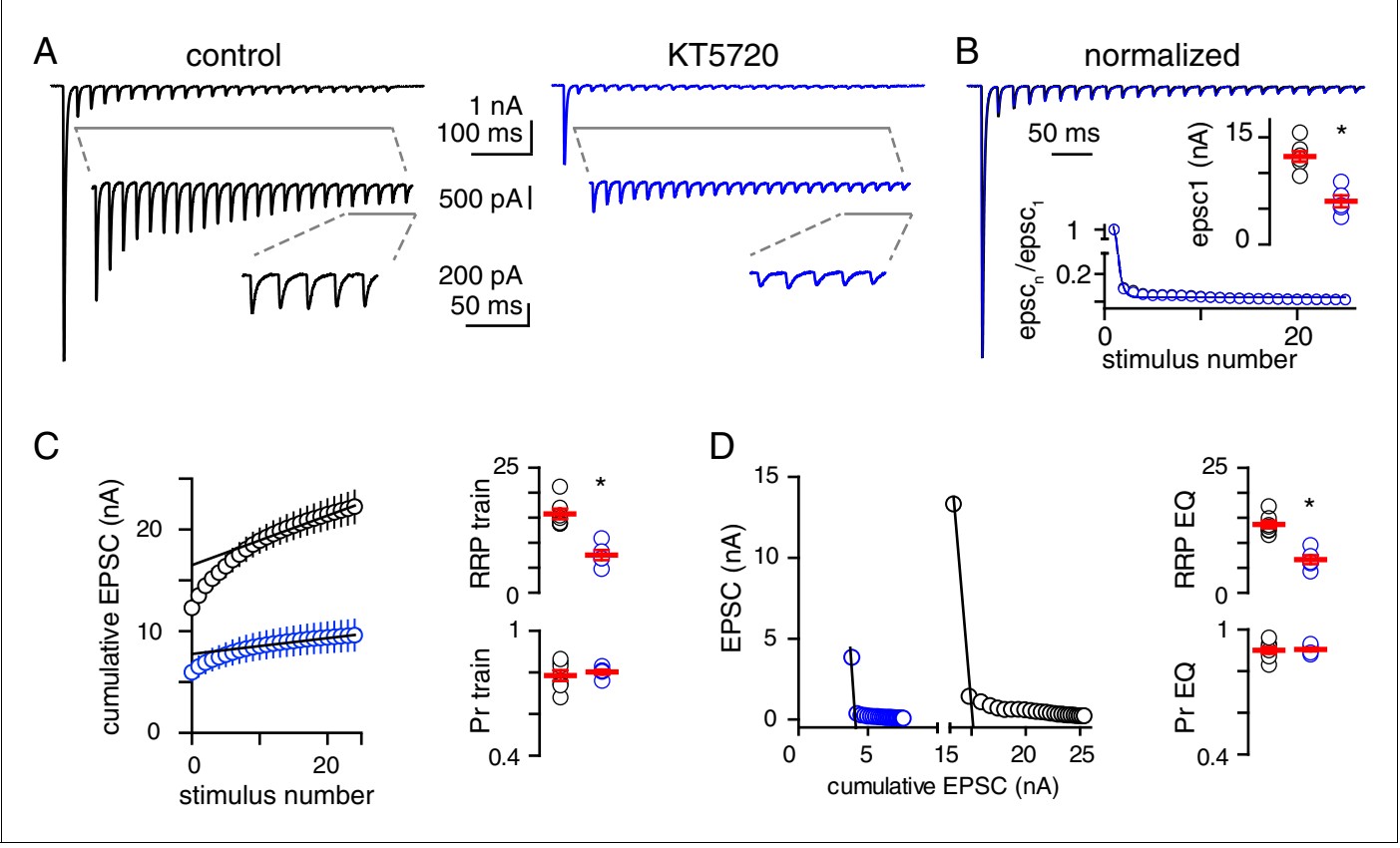

**Figure 5.** PKA inhibition reduces the size of the RRP. (A) Representative EPSCs recorded in response to CF stimulation at 50 Hz for 500 ms in control (black) and KT5720-treated (blue) slices. To relieve receptor saturation, 3 mM KYN was added in a recording solution with 2.5 mM $Ca^{2+}$; $V_m = -60$ mV). (B) Superimposed train responses from control (black) and KT5720-treated (blue) slices, normalized to the first EPSC. Inset$_1$: Average EPSC1 amplitude in control and KT5720-treated slices. Inset$_2$: Normalized EPSC amplitude plotted as a function of stimulus number. The time constants were similar between control (23.3 ± 0.6 ms, n = 7) and KT5720-treated slices (22.9 ± 0.4 ms, n = 5, p=0.6, unpaired t-test). (C, left) Cumulative EPSC amplitude plotted as a function of stimulus number for control (black) and KT5720-treated (blue) slices. A line was fit to the final 5 EPSCs in each condition. (C, right) The RRP$_{train}$ (15.8 ± 1.0 nA and 7.5 ± 1.0 nA, n = 7 and 5; p=0.0002, unpaired t-test) and Pr$_{train}$ (0.78 ± 0.03 and 0.80 ± 0.05, n = 7 and 5; p=0.61, unpaired t-test) were calculated from this plot. (D, left) Representative plots of EPSC amplitude versus the cumulative EPSC in control (black) and KT5720-treated (blue) recording with linear regressions to the initial portion of each data set. (D, right) The RRP$_{eq}$ (13.7 ± 0.7 nA and 6.6 ± 0.9 nA, n = 7 and 5; p=0.0001, unpaired t-test) and Pr$_{eq}$ (0.90 ± 0.02 and 0.90 ± 0.01, n = 7 and 5; p=0.85, unpaired t-test) were calculated from this plot.

DOI: https://doi.org/10.7554/eLife.47434.013

transmitter release (*Hosaka et al., 1999*; *Hilfiker et al., 2005*). Synapsins are the most abundant family of synaptic vesicle-associated phosphoproteins and are encoded by three distinct genes (synapsin I, II, III). At excitatory synapses, they are typically associated with regulation of the reserve pool of synaptic vesicles because complete loss of synapsin I, II and III does not alter transmission evoked by single stimuli but rather alters the rate of depression during repetitive stimulation (*Gitler et al., 2004a*; *Gitler et al., 2004b*; *Gitler et al., 2008*; *Vasileva et al., 2012*). Synapsin-dependent regulation of the RRP has been documented at inhibitory synapses (*Baldelli et al., 2007*), suggesting varied roles that depend on the types of synapses or preparations studied (*Greengard et al., 1993*; *Rosahl et al., 1995*; *Esser et al., 1998*; *Hilfiker et al., 1998*; *Hosaka and Südhof, 1998*; *Song and Augustine, 2015*).

Thus, we tested PKA regulation of MVR in mice in which all three synapsin genes were deleted (TKO) or in a line of matching heterozygous mice (het) derived in parallel (*Gitler et al., 2004a*). In 2.5 mM $Ca^{2+}$, CF-PC EPSCs from het and TKO mice had similar kinetics, short-term plasticity, and recovered from depression with a time course similar to their wildtype counterparts (data not shown and *Figure 6—figure supplement 1A*). Our estimates of Pr at 10 ms ISI, when little replenishment is assumed to have taken place, were similar to that in wildtype recordings (0.92 ± 0.01 and

0.93 ± 0.01 at 10 ms ISI, n = 10 and 8, p=0.4). Likewise, high frequency train stimulation (50 Hz, 25 stimuli) resulted in marked depression that was similar to wildtype recordings in both genotypes *Figure 6—figure supplement 1B*; normalized train amplitudes plotted versus stimulus number for WT, het and TKO recordings were all fit with the same exponential decay, p=0.68; extra sum-of-squares t-test, $R^2$ = 0.99 for all datasets; ratio of last EPSC to first for WT: 0.017 ± 0.004; het: 0.010 ± 0.001; TKO: 0.013 ± 0.002, 1-way ANOVA, p=0.33, n = 6 each). However, peak EPSC amplitudes from TKO mice were 45% smaller compared to their het littermates (het: 13.5 ± 0.98 nA; TKO: 7.8 ± 0.59 nA, n = 6 each, p=0006). We estimated the size of the RRP and found that, on average, the RRP from TKO mice was about half the size as in het littermates without a change in estimated Pr. Both methods of calculating Pr ($Pr_{train}$ and $Pr_{EQ}$) generated measures that were similar to one another and to Pr measures from wildtype mice (*Figure 6—figure supplement 1C*). Together these results show that neurotransmitter release in TKO mice mimics inhibition of PKA signaling: reduced RRP with no change in Pr (compare with *Figure 4C* and *Figure 5*). In the next set of experiments, we tested the idea that that PKA controls MVR via synapsin-dependent control of the RRP.

First, we recorded in low Pr conditions, when UVR predominates (0.5 mM $Ca^{2+}$), and showed that 6-Bnz potentiated EPSCs in het mice by 34% (from 0.91 ± 0.12 nA to 1.22 ± 16 nA, n = 7, p=0.008, paired t-test) without a change in PPR (*Figure 6Ai*; from 1.46 ± 0.07 to 1.42 ± 0.06, p=0.54), similar to EPSCs from wildtype slices. However, 6-Bnz had no effect on either the amplitude or PPR of CF-EPSCs in TKO mice (*Figure 6Aii*). Consistent with wildtype mice (*Figure 1*), EPSCs enhanced by Bnz-cAMP in het mice exhibited reduced inhibition by KYN whereas KYN inhibition was unchanged by 6-Bnz in TKO littermates (*Figure 6Bi*). As a control, inhibition by NBQX was similar in all conditions (*Figure 6Bii*). Thus, PKA-induced MVR was absent in synapsin TKO mice.

In conditions when MVR predominates (2.5 mM $Ca^{2+}$), incubating slices with the PKA inhibitor KT5720 (1 μM) increased KYN (1 mM) inhibition of EPSCs in het mice (*Figure 6Ci*), reminiscent of the results in wildtype mice (*Figure 4*). Interestingly, the baseline KYN inhibition in TKO mice was similar to het or wildtype responses when PKA signaling was inhibited, however KT5720 had no additional effect on KYN inhibition in slices from TKO mice (*Figure 6Ci*) while inhibition by NBQX was similar in all conditions (*Figure 6Cii*). As with all other manipulations of PKA signaling, KT5720 did not alter the PPR of EPSCs in het mice (*Figure 6Ciii*) or in TKO mice (*Figure 6Civ*). Thus, suppression of MVR by PKA inhibition was absent in synapsin TKO mice. Altogether these data suggest that synapsins may be the principal targets of PKA at CF synapses and that this family of proteins is required for bidirectional regulation of MVR via regulation of the RRP.

## RRP regulates MVR independent of Pr

Our results show PKA-dependent regulation of the RRP bidirectionally transforms the release mode at CF synapses (UVR ⇔ MVR) without affecting Pr. The simplest explanation for our results is that the RRP is a functional measure of the number of *release-competent* vesicles per active zone, that is under the control of synapsins, and constrains the number of vesicles released (MVR). Surprisingly, this fundamental synaptic parameter has not been explored, leading to the exclusive focus on Pr as the principal regulator of UVR ⇔ MVR (reviewed in *Rudolph et al., 2015*). To understand how RRP size affects MVR across the range of release probabilities, we used a classic model that incorporates facilitation and depression to estimate vesicle release (*Dittman et al., 2000*) and added a reaction step to account for the interaction of vesicles with docking sites. We use common terminology and abbreviations to be consistent with a recently proposed two-step model of release (*Miki et al., 2016*). For simplicity, the size of the RRP is shown to be dependent on synapsin that may also function to assist in the priming of already-docked vesicles (*Figure 7A*). In our scheme, the phosphorylation state of synapsin is critical to controlling the probability of site occupancy and the RRP size upstream from the probability that a competent vesicle will undergo fusion. Vesicles in this RRP are *only then* subject to calcium-driven reactions (Ca-X) that drive facilitation as well as recovery from depression, parameters that dictate short-term plasticity.

We simulated each active zone with a single or multiple sites where vesicles dock for subsequent priming and release (*Pulido and Marty, 2017*). We used our model to determine how the number of docking sites affects the number of released vesicles ($V_{released}$) within an active zone where vesicles are subject to high and low probability of release once competent ($P_{succ}$). Not surprisingly, a one-site active zone can release, at most, one vesicle because $P_{succ} * P_{occ} *$ number of release sites ≤ 1 (*Figure 7Bi*). However, this relationship steepens and is best described by a hyperbolic

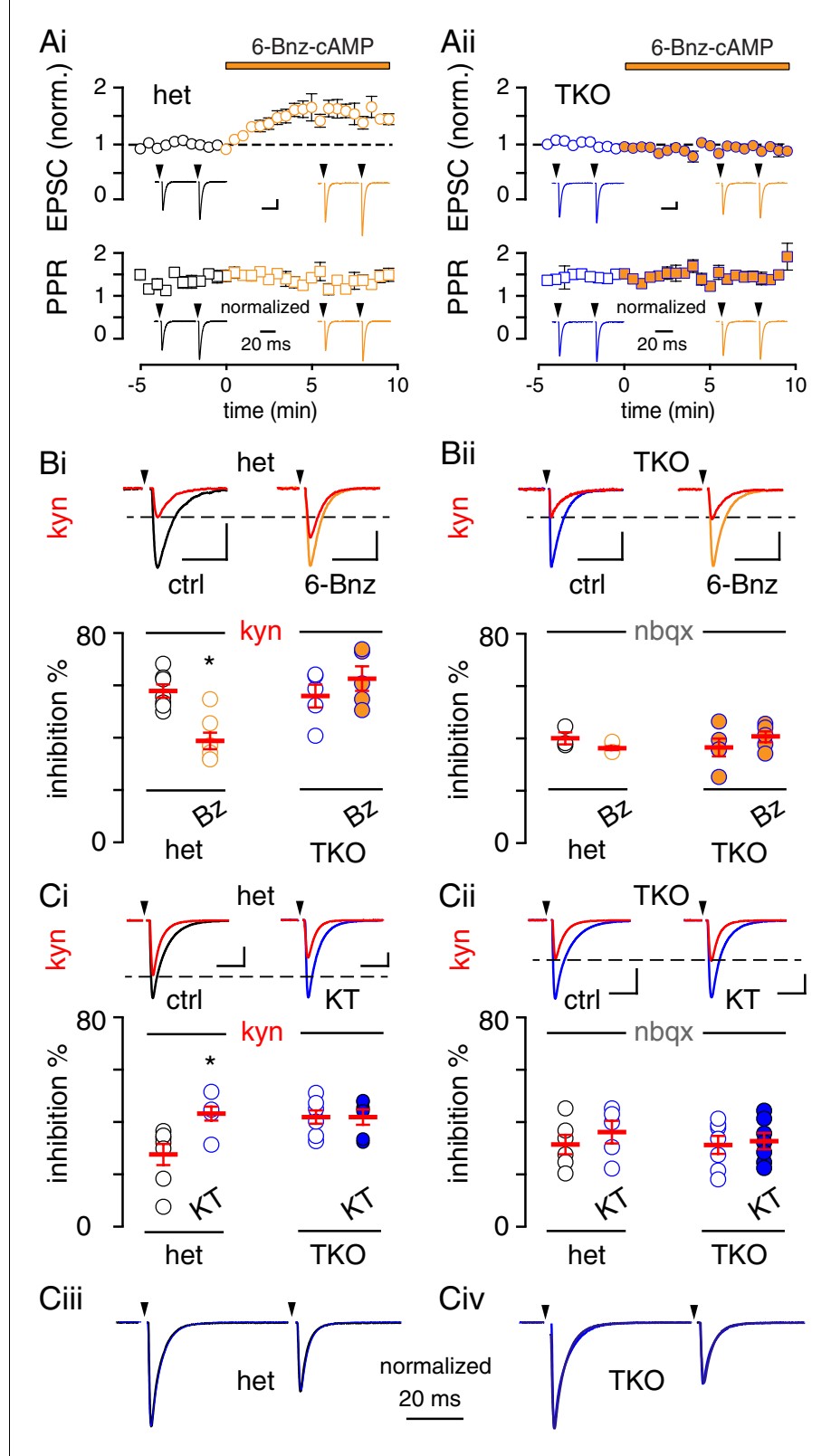

**Figure 6.** PKA-mediated regulation of MVR is absent in synapsin TKO mice. (**A**) Time course of CF-PC EPSC amplitude (top: normalized, circles) and paired pulse ratio (bottom: PPR with an inter-stimulus interval = 50 ms, squares) following bath application of 6-Bnz (orange) from slices of synapsin triple het (**Ai**, black) or TKO (**Aii**, brown) mice recorded in 0.5 mM $Ca^{2+}$. Insets: representative and normalized traces show the effects on amplitude

*Figure 6 continued on next page*

*Figure 6 continued*

and lack of change in PPR. Het-slices: EPSC from 0.91 ± 0.12 nA to 1.22 ± 16 nA, n = 7; p=0.008, paired t-test. PPR from 1.46 ± 0.07 to 1.42 ± 0.06; n = 7, p=0.54. TKO-slices EPSC from 0.96 ± 0.21 nA to 0.89 ± 0.22 nA, n = 6; p=0.82, paired t-test. PPR from 1.42 ± 0.09 to 1.49 ± 0.11, n = 6; p=0.06. Scale bars: 200 pA, 10 ms. (B) Top, blockade of EPSCs by KYN (red) in control (black) and 6-Bnz (orange) in slices from synapsin triple het (Bi, black) or TKO (Bii, blue) mice. Recordings were made in 0.5 mM $Ca^{2+}$. Bottom, 6-Bnz reduced the KYN (0.25 mM) block in het slices (control: 57.9 ± 2.5%, n = 7; 6-Bnz: 38.2 ± 3.2, n = 7; p=0.003) but not in slices from TKO (control: 55.9 ± 4.4%, n = 5; 6-Bnz: 62.6 ± 4.7%, n = 5; p=0.56, ANOVA with Holm-Sidak post-test). NBQX was similar in all conditions (het slices: control: 40.7 ± 2.4% and 6-Bnz: 36.9 ± 0.8, n = 3 each; p=0.87. TKO slices: control: 37.2 ± 3.4% and 6-Bnz: 41.2 ± 2.2, n = 5 each; p=0.84. Scale bars: 1 nA, 10 ms (hets) and 500 pA, 10 ms (TKO). (C) Top, blockade of EPSCs by KYN (red) in control (black) and KT 5720-treated slices (blue) from synapsin triple het (Ci) or TKO (Cii) mice. Recordings were made in 2.5 mM $Ca^{2+}$. Bottom, KT5720 increased the KYN (1 mM) block in het slices (control: 27.6 ± 4.0% and KT5720: 43.2 ± 2.7, n = 6 and 7, p=0.02), but there was no difference in KYN block in slices from TKO mice (control: 41.8 ± 2.5% and KT5720: 41.8 ± 2.9%, n = 6 each; p=0.99, 2-way ANOVA). NBQX blocked EPSCs to a similar extent in all conditions (het slices: control: 31.3 ± 3.6% and KT5720: 36.1 ± 4.3, n = 5 and 6, p=0.93; TKO slices: control: 31.2 ± 3.4% and KT5720: 32.7 ± 3.1, n = 7 each; p=0.4, ANOVA). Scale bars: 2 nA, 10 ms. (Ciii and Civ) KT5720 incubation had no effect on PPR in slices from either het mice (control: 0.67 ± 0.02, n = 11; KT5720: 0.62 ± 0.02, n = 8; p=0.75) or in TKO mice (control: 0.65 ± 0.04, n = 6; KT5720: 0.64 ± 0.03, n = 8; p=0.96).

DOI: https://doi.org/10.7554/eLife.47434.014

The following figure supplement is available for figure 6:

**Figure supplement 1.** Synaptic release in TKO mice mimics inhibition of PKA signaling, with a smaller RRP but no change in Pr.

DOI: https://doi.org/10.7554/eLife.47434.015

---

paraboloid (resembles a Pringles chip) as release site numbers increase, making UVR (gray fraction) less frequent (*Figure 7Bii, Biii*). We found that the probability of MVR (orange/red fraction when $V_{released}$ >1,>2, etc.) increases logarithmically with the number of docking sites per active zone (*Figure 7Biv*). Notably, CFs are reported to have, on average, 7–8 docked vesicles (*Xu-Friedman et al., 2001*), a measure that may be analogous to the total number of sites in our model ($N_T$).

Most importantly, our model reconciles how UVR and MVR can co-exist under high or low conditions of $P_{succ}$. At high $P_{succ}$ and $P_{occ}$, equivalent to 2.5 mM $Ca^{2+}$, an average of 4.5 vesicles are released at an active zone with seven sites (*Figure 7C*, point a). Reducing only $P_{occ}$ (i.e. KT5720 incubation) decreases the average number of released vesicles to <1 (UVR) without affecting the PPR (*Figure 7C*, point b). At low $P_{succ}$ and $P_{occ}$, equivalent to 0.5 mM $Ca^{2+}$, the average number of vesicles is also <1 (UVR) but with a PPR = 1.2, as expected for low $P_{succ}$ (*Figure 7C*, point c). Increasing $P_{occ}$ (i.e. 6-Bnz treatment) raises the number of vesicles released (to 2.2 vesicles) without changing PPR (*Figure 7C*, point d). Thus, a model that incorporates PKA-mediated control of the RRP is sufficient to recapitulate our experimental results and reconcile those at other synapses. These results show that PKA-mediated regulation of RRP is sufficient to generate a bi-directional shift in release mode (from MVR→UVR or UVR→MVR) without changing $P_{succ}$, thus illustrating that high Pr is not required for MVR at CF synapses where this release mode predominates.

## Regulation of vesicular release mode at simple synapses

PKA-mediated changes in the size of the RRP could contribute to heterogeneity in MVR across CNS synapses. We thus sought to evaluate the generality of our model by testing cAMP/PKA regulation of MVR at a more conventional excitatory synapse that exhibits a diversity of release properties. The regular arrangement of cerebellar parallel fibers (PFs) allows isolation of individual PF-molecular layer interneuron (MLI) synapses using minimal stimulation (*Nahir and Jahr, 2013*; *Malagon et al., 2016*). PF axons rarely form multiple contacts with an individual postsynaptic cell (*Napper and Harvey, 1988*) and the maximum number of released vesicles indicates the presence of several docking sites, ranging between 2 and 10 (*Malagon et al., 2016*; *Pulido and Marty, 2017*). Assessing synaptic responses to train stimulation of single PF synapses revealed a diversity of UVR or MVR across a large range of Prs (*Nahir and Jahr, 2013*; *Malagon et al., 2016*). To test whether cAMP/PKA-dependent regulation of MVR occurs at PF synapses, we delivered a 50 Hz train of five stimuli

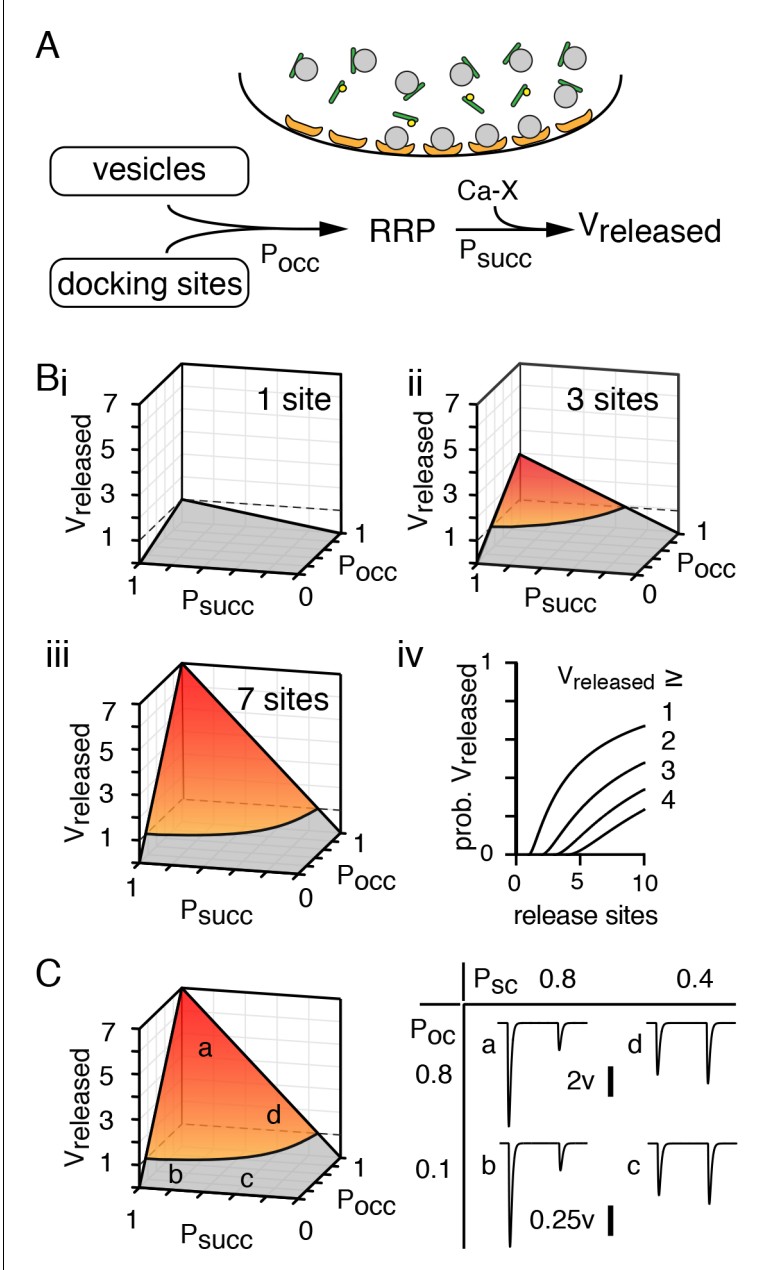

**Figure 7.** Synapsin-mediated regulation of the RRP controls MVR independent of Pr. (**A**) Cartoon and schematic of short-term plasticity model showing how synapsin controls the number of release-competent sites per active zone. Synaptic vesicles (gray) are bound by synapsin (green) restricting the size of the readily-releasable pool of vesicles that is limited by the number of release sites (orange). Synapsin phosphorylation (yellow) allows vesicles to become part of the RRP upstream from subsequent $Ca^{2+}$-dependent facilitation and recovery from depression as in *Dittman et al. (2000)* that dictate $V_{released}$. Although, we have illustrated that synapsin phosphorylation controls vesicle docking, its influence can also be exerted by priming already docked vesicles. (**Bi - iii**) The number of vesicles released per active zone ($V_{released}$) is shown with increasing number of release sites per active zone and is the product of the probability of site occupancy ($P_{occ}$), the probability that a competent vesicle will release ($P_{succ} = F*D$), and number of release sites ($N_T$). The product of these parameters steepens with increasing number of release sites. (**Biv**) The probability of observing more than 1–4 vesicles follow a logarithmic function (see Materials and methods). (**C**) $V_{released}$ and simulated EPSCs in four conditions (points *a - d*) show that the number of vesicles released ($V_{released}$) does not require a high probability that a competent vesicle will release ($P_{succ}$) and that short-term plasticity does not predict release mode.

DOI: https://doi.org/10.7554/eLife.47434.016

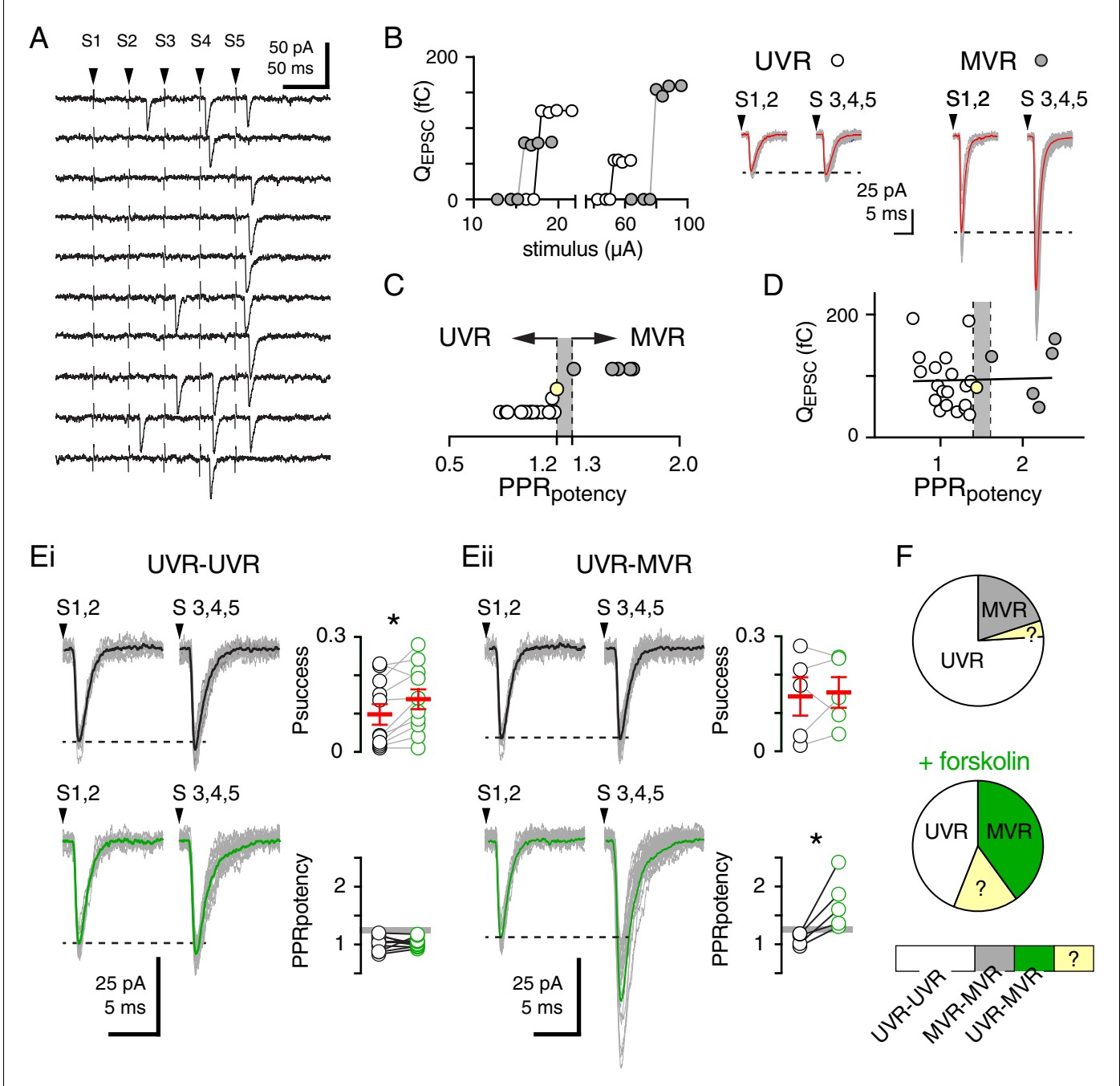

**Figure 8.** Fsk increases MVR at PF-MLI synapses independent of Pr. (A) Unitary responses following five parallel fiber stimuli (S1– S5) delivered at 50 Hz. (B left) Individual parallel fibers were identified by a sharp threshold between failures and successes (S1 and S2) with increasing stimulus intensity. Four representative experiments demonstrating that all-or-none responses at UVR (open circles) or MVR (grayed circles) synapses. Classification as UVR does not depend on stimulus intensity or synaptic charge (also see (D)). (B middle) The potency of UVR synapses in final three stimuli (S3, 4, 5) was similar to that of the first two stimuli (S1, 2). (B right) The potency of MVR synapses increased later in the train. Superimposed individual successful responses are shown in gray and average responses are shown in red. (C) The ratio of the potencies of S3, 4, 5 to S1, 2 ($PPR_{potency}$) was used to classify synapses as UVR (0.8–1.2) or MVR (>1.3). Synapses with $PPR_{potency}$ between 1.2 and 1.3 (gray area) were excluded from further analysis. (D) Lack of correlation between release mode ($PPR_{potency}$) and the size of the response to S1,2 ($R^2 = 0.001$; p=0.9). (Ei) In a subset of cells, fsk (50 µM; green) increased the probability of success ($P_{success}$ of S1-2 stimuli: $0.10 \pm 0.03$ vs. $0.14 \pm 0.03$; p=0.01, paired t-test, n = 11) of UVR synapses with no effect on release mode ($PPR_{potency}$: $1.03 \pm 0.04$ vs $1.03 \pm 0.2$; p=0.9, paired t-test, n = 11). Individual responses are shown in gray and average responses before or following fsk are shown in black or green, respectively. (Eii) In another subset of cells, fsk increased PPR potency from $1.1 \pm 0.05$ to $1.7 \pm 0.2$ (p=0.03, paired t-test, n = 5), changing their classification from UVR to MVR without altering $P_{success}$ ($0.14 \pm 0.05$ vs. $0.15 \pm 0.04$, p=0.74, paired t-test, n = 5).
*Figure 8 continued on next page*

*Figure 8 continued*

Individual responses are shown in gray and average responses before or following fsk are shown in black or green, respectively. (F) Summary data showing that in control conditions (top pie chart), 19/25 synapses were classified as UVR, 5/25 were MVR, and 1/25 could not be classified. Following fsk (bottom pie chart), 11/25 synapses were UVR, 10/25 were MVR, and four could not be classified. Grouped data (bottom) showing the classification of synaptic responses before and after fsk treatment: 11/25 were unchanged from UVR (UVR-UVR), 5/25 were converted from UVR to MVR, five were unchanged from MVR (MVR-MVR), and four could not be classified in one or both conditions because PPR$_{potency}$ was neither <1.2 nor >1.3.

DOI: https://doi.org/10.7554/eLife.47434.017

(*Figure 8A*) and defined successful single synaptic responses as those that showed (a) short response latency, (b) consistent EPSC kinetics, (c) clear differentiation between success and failures, and (d) all-or-none responses (*Figure 8A and B*). As previously described, the paired-pulse ratio of potency (PPR$_{pot}$; potency = average amplitude of successful EPSCs following a given stimulus) was used to classify individual synapses as UVR if PPR$_{pot}$ was between 0.8 and 1.2, or as MVR if PPR$_{pot}$ >1.3 (*Nahir and Jahr, 2013*) (*Figure 8B and C*). Using these criteria, 75% of responses (19 of 25; PPR$_{pot}$ = 1.0 ± 0.03; P$_{success}$ = 0.12 ± 0.02) were categorized as UVR and 20% of responses (5 of 25) were considered MVR (PPR$_{pot}$ = 1.6 ± 0.05; P$_{success}$ = 0.13 ± 0.05). Importantly, the amplitude of EPSCs did not predict whether responses were classified as UVR or MVR, suggesting that variability in the number of postsynaptic receptors is equivalent between sites with different release modes (*Figure 8D*). Furthermore, P$_{succ}$ across stimuli was low and also did not predict UVR or MVR at individual synapses (see below; *Nahir and Jahr, 2013*).

We next tested whether enhancing cAMP was sufficient to enhance MVR independent of Pr. As at CFs (see *Figure 3*), PF synapses that showed pronounced MVR (PPR$_{pot}$ >1.3), fsk (50 μM) did not change either the P$_{succ}$ or PPR$_{pot}$ (n = 5; p=0.57 and 0.77; data not shown). In contrast, fsk slightly increased P$_{succ}$ (0.10 ± 0.03 to 0.14 ± 0.03, p=0.01) without changes to PPR$_{pot}$ (1.03 ± 0.04 to 1.03 ± 0.02, p=0.88) in 11 of 19 UVR synapses, a result that is qualitatively similar to cAMP/PKA potentiation attributed to increases in Pr (*Salin et al., 1996*; *Bender et al., 2009*) (*Figure 8Ei*). Interestingly, fsk increased PPR$_{pot}$ at five additional UVR synapses (from 1.1 ± 0.04 to 1.7 ± 0.2, p=0.03) that failed to exhibit changes in P$_{succ}$ during the train (0.14 ± 0.05 to 0.15 ± 0.04, p=0.74; *Figure 8Eii*). This suggests that approximately 25% of PF synapses can alter release mode from UVR to MVR in response to potentiation of cAMP/PKA signaling (*Figure 8F*). Altogether these results show that cAMP/PKA signaling can enhance MVR without altering Pr at PF synapses as well as CF synapses, and that this signaling pathway can also facilitate presynaptic function by enhancing Pr at PF synapses. Since PF synapses with fsk-induced enhanced Pr did not display MVR, we conclude that cAMP/PKA signaling primarily elevates MVR via regulation of the RRP.

## Discussion

Our results show that the number of glutamate-filled vesicles released at individual CF terminals with each action potential is subject to bidirectional modification of the RRP. Altered cAMP/PKA activity changed the EPSC inhibition by a LAA, a result that indicates a change in the synaptic glutamate concentration and thus the number of vesicles released per active zone. Surprisingly, changes in cAMP/PKA activity did not affect Pr, as quantified by four separate measures, but rather altered the size of the RRP. We found that PKA-dependent signaling can shift the balance between MVR and UVR, but is maximally active in 2.5 mM Ca$^{2+}$ at CFs. Furthermore, regulation of release mode by PKA requires synapsins. Our data is explained by a model where release site occupancy or vesicle competency depend on the size of the RRP, upstream of Pr. These results propose a new function for the RRP and molecules, like the synapsins, that may be applicable across central synapses.

### Vesicle release probability and MVR

The original hypothesis that Pr exclusively dictates the propensity for MVR makes two assumptions: (1) that each active zone contains multiple docking sites, and (2) that each docking site (sometimes referred to as a release site) is filled at rest, thus holding a primed vesicle that is competent for release. Several methods have addressed the first assumption by estimating the number of docking sites at each active zone. Anatomical studies can quantify the number of docked vesicles with exquisite detail (*Imig et al., 2014*; *Molnár et al., 2016*) whereas imaging of presynaptic proteins or

vesicle cycling can convey vesicle pool dynamics within active zones (*Ryan et al., 1997*; *Zenisek et al., 2000*; *Indriati et al., 2013*; *Midorikawa and Sakaba, 2015*). Functional measures of vesicle release based on postsynaptic amplitude fluctuations can integrate measures of docking sites and released vesicles (*Pulido et al., 2015*); reviewed in *Pulido and Marty (2017)*. Despite the distinct caveats of the varied approaches, these studies largely agree that each active zone has multiple docking sites (or release sites), albeit with a large intersynaptic heterogeneity. Our approach using inhibition of EPSCs by a LAA likewise demonstrates that each active zone contains multiple release sites. While our approach provides only a relative measure of neurotransmitter concentration per active zone, it is one of few functional assays that delineates changes in MVR at synapses comprised of multiple active zones and thus it is indispensable for interpreting the effects of PKA modulation that may be undetectable by other assays. Together with prior studies, our results support the assumption that each active zone contains multiple docking sites.

However, many studies provide converging evidence against the second assumption that every release site is occupied at rest and the analogous assumption that an anatomically-defined docked vesicle is release competent. For example, at the crayfish neuromuscular junction release site refractoriness was necessary to accurately model experimental data (*Pan and Zucker, 2009*). This is conceptually similar to a docking site probability of <1 as has been proposed at many inhibitory and excitatory synapses (*von Gersdorff et al., 1996*; *Rizzoli and Betz, 2004*; *Neher, 2010*; *Pulido et al., 2015*; *Miki et al., 2016*; *Neher and Brose, 2018*). Evidence against the second assumption argues that factors beyond Pr must also contribute to MVR.

We propose a model that explains the propensity for MVR at low Pr synapses and UVR at high Pr synapses by incorporating static docking site number with dynamic docking site occupancy and vesicle release probability. We have modeled CFs as having seven docking sites per active zone based on anatomically-defined docked vesicles (*Xu-Friedman et al., 2001*) with a variable degree of docking site occupancy. While it would be functionally equivalent for the number of active docking sites to serve as the dynamic variable with a constant occupancy, data from synapsin-TKO shows no difference in the number of docked vesicles (*Gitler et al., 2004a*). If we consider CFs that have 300 active zones (*Llinas et al., 1969*); and thus ~2100 sites), the probability of CF docking/release site occupancy is ~0.7, calculated from an estimate of 1400 functional docking sites (*Foster and Regehr, 2004*). As shown in *Figure 7C*, altering docking site occupancy is sufficient to shift the number of vesicles released per active zone without changing Pr ($P_{succ}$), replicating our experimental data.

Activation of the cAMP cascade can alter synaptic transmission at PF-PC synapses by increasing the conductance of GluA3-containing AMPARs (*Gutierrez-Castellanos et al., 2017*) but see *Chen and Regehr (1997)*. Though our initial finding that activation of cAMP/PKA enhanced EPSC amplitude without changes in short-term plasticity or CV is consistent with modulation of AMPAR function, several lines of evidence argue against the possibility. First, neither activation nor inhibition of PKA altered the amplitude or kinetics of quantal-like aEPSCs. Second, bath application of a PKA activator potentiated EPSCs even when a PKA inhibitory peptide was included in the pipette solution. Third, the effects of PKA activation were dependent on the external $[Ca^{2+}]$, as expected for a presynaptic process. Fourth, PKA modulation was associated with changes in the block by a LAA. Finally, activation of cAMP signaling did not enhance synaptic transmission in Synapsin TKOs. Taken together, these findings argue that modulation of the RRP, and not AMPARs, mediate the effects of PKA.

## Synapsins, RRP, and synaptic transmission

Synapsins were initially proposed to control transitions of synaptic vesicles from the reserve pool to the RRP through phosphorylation-dependent interactions with actin (*Greengard et al., 1993*; *Ceccaldi et al., 1995*; *Hosaka et al., 1999*; *Chi et al., 2001*) and were later shown to be fundamental to post-docking steps of exocytosis (*Rosahl et al., 1995*; *Hilfiker et al., 1998*; *Sun et al., 2006*). Although synapsins constitute one of the most abundant (*Wilhelm et al., 2014*) and highly studied families of presynaptic proteins, a consensus view of their role in synaptic transmission may be hampered by subtype redundancy and potential for cell-type specific functions (for example see *Gitler et al., 2004a*; *Song and Augustine, 2016*). Motivated by experiments at the squid giant terminal (*Llinás et al., 1991*; *Hilfiker et al., 2001*), we focused on regulation of neurotransmitter release by PKA-synapsin with the caveat that other kinases may also act similarly (i.e. *Llinás et al., 1991*).

Interestingly, we found that cAMP/PKA stimulation increased release only under low extracellular $Ca^{2+}$ whereas, the effects of cAMP/PKA activation were occluded in elevated $Ca^{2+}$. It is well known that robust $Ca^{2+}$ influx is sufficient to stimulate adenylyl cyclases (ACs) to generate cAMP (*Salin et al., 1996*; *Wong et al., 1999*; *Moulder et al., 2008*). Sufficient $Ca^{2+}$ influx in 2.5 mM extracellular $Ca^{2+}$ may thus allow stimulation of this pathway at CF synapses. This could reflect the short diffusional distance between $Ca^{2+}$ channels and release sites and/or high $Ca^{2+}$ channels density at depressing presynaptic terminals (*Rozov et al., 2001*). $Ca^{2+}$-dependent ACs and PKA may also be compartmentalized in distinct subcellular domains providing additional mechanisms to regulate the $Ca^{2+}$ sensitivity of this pathway (*Steinberg and Brunton, 2001*). We speculate that the $Ca^{2+}$ dependence of cAMP/PKA-synapsin signaling at CF synapses contributes to the basal size of the RRP that is known to change in response to extracellular $Ca^{2+}$ (compare *Figures 2* and *5*; *Schneggenburger et al., 1999*; *Neher and Sakaba, 2008*; *Thanawala and Regehr, 2013*). An intriguing possibility is whether external $Ca^{2+}$ affects AC/PKA activity at rest, independent of $Ca^{2+}$ influx. In cultured neurons, a membrane bound $Ca^{2+}$ sensitive receptor (CaSR) mediates increased mEPSC release when external $Ca^{2+}$ is raised, independent of both $Ca^{2+}$ channels and internal $Ca^{2+}$ chelation (*Vyleta and Smith, 2011*). Diversity in $Ca^{2+}$ homeostasis, ACs and PKA may thus contribute to the variability in PKA and synapsin-dependent regulation of release across synapses.

## Incorporating synapsins into current release models

Recent release models have proposed that vesicles transition between sequential states: from loose to tighter tethering with release machinery followed by $Ca^{2+}$-triggered fusion (*Neher and Brose, 2018*), which may be akin to transitions from replacement sites to docking sites prior to fusion (*Miki et al., 2016*). These transitions are likely regulated by intracellular $Ca^{2+}$ and thus, may occur too rapidly to discriminate using electrophysiological methods (*Neher and Brose, 2018*). Therefore, estimates of release probability include the equilibrium between states. Our results support these models, assuming that molecules such as synapsins contribute to the molecular machinery upstream of these transitions, since PKA-dependent synapsin signaling changes the size of the RRP without affecting the probability of $Ca^{2+}$-triggered fusion. Recently, synapsin was proposed to form a distinct liquid phase that captures synaptic vesicles and is rapidly disassembled upon phosphorylation by CamKII (*Milovanovic et al., 2018*). We speculate that PKA phosphorylation triggers a similar mechanism that allows vesicles to enter the loose/replacement sites.

# Materials and methods

**Key resources table**

| Reagent type (species) or resource | Designation | Source or reference | Identifiers | Additional information |
|---|---|---|---|---|
| Gnetic reagent (*M. musculus*) | wildtype; WT; control | Jackson Laboratories | RRID:IMSR_JAX:000664 | |
| Genetic reagent (*M. musculus*) | synapsin triple knockout, TKO | MMRC | RRID:MMRRC_041434-JAX | |
| Genetic reagent (*M. musculus*) | synapsin triple het; het | this paper | | WT x TKO cross |
| Peptide, recombinant protein | PKA inhibitory fragment (6-22) amide, PKi | Tocris | Cat#: 1904; CAS: 121932-06-7 | |
| Chemical compound, drug | 6-Bnz-cAMP; 6-Bnz | BioLog via Axxora | Cat#: B009; CAS: 30275-80-0 | |
| Chemical compound, drug | 8-Br-cAMPs; 8-Br | Santa Cruz | Cat#: B009; CAS: 30275-80-0 | |
| Chemical compound, drug | forskolin; fsk | HelloBio | Cat#: HB1348; CAS: 66575-29-9 | |
| Chemical compound, drug | KT5720; KT | Tocris | Cat#: 1288; CAS: 108068-98-0 | |
| Chemical compound, drug | kynurenic acid, KYN | Abcam | Cat#: ab120256; CAS: 494-27-3 | |

*Continued on next page*

*Continued*

| Reagent type (species) or resource | Designation | Source or reference | Identifiers | Additional information |
|---|---|---|---|---|
| Chemical compound, drug | NBQX | Abcam | Cat#: ab120045; CAS: 118876-58-7 | |
| Chemical compound, drug | Picrotoxin | Abcam | Cat#: ab120315; CAS: 124-87-8 | |
| Chemical compound, drug | QX-314 | Abcam | Cat#: ab120118; CAS: 5369-03-9 | |
| Software, algorithm | Axograph X, version 1.5.4 | AxoGraph Scientific | https://axograph.com/ | |
| Software, algorithm | Mathematica 11 | Wolfram | http://www.wolfram.com/mathematica/ | |
| Software, algorithm | pCLAMP 10 | Molecular Devices | https://www.moleculardevices.com/ | |
| Software, algorithm | Prism | Graphpad | https://www.graphpad.com/ | |

Further information and requests for resources and reagents should be directed to and will be fulfilled by the lead Contacts Linda Wadiche (lwadiche@uab.edu) or Jacques Wadiche (jwadiche@uab.edu).

## Experimental model and subject details

We used male and female mice aged P12-P18. Only mice with fully open eyes were used and only PCs with a single CF were included in our analysis. C57BL/6J (RRID: IMSR_JAX:000664) and B6:129-*Syn2* $^{tm1Pggd}$, *Syn3* $^{tm1Pggd}$, *Syn1* $^{tm1Pggd}$/Mmjax (RRID: MMRRC_041434-JAX), referred to as TKOs, were purchased from Jackson Laboratories (Bar Harbor, ME) and maintained in our colony in standard housing in a 12:12 hr light: dark cycle. Mice heterozygous for each synapsin mutation (triple hets and referred to as hets) were generated by breeding TKO males to C57 females, and offspring were compared to TKOs with the same sire. TKOs and triple hets were fed Picolab Rodent Diet 20 (LabDiet, St. Louis, MO). All experiments were conducted through protocols approved by the Institutional Animal Care and Use Committee of the University of Alabama at Birmingham under protocol #08767.

## Slice preparation

Mice were deeply anesthetized via isoflurane (VetOne, Boise, ID) and rapidly decapitated. The cerebellum was dissected into ice cold cutting solution containing (in mM): 125 NaCl, 2.5 KCl, 1.0 $NaH_2PO_4$, 26.2 $NaHCO_3$, 11 glucose, 0.5 $CaCl_2$, and 3.3 $MgCl_2$ (for PC recordings) or 110 choline chloride, 2.5 KCl, 1.25 $NaH_2PO_4$, 25 $NaHCO_3$, 25 glucose, 11.5 sodium ascorbate, three sodium pyruvate, 0.5 $CaCl_2$, and 7.0 $MgCl_2$ (for MLI recordings), bubbled with 95% $O_2$, 5% $CO_2$. The cerebellum was mounted on an agar block on the stage of a vibrating microtome (VT1200S, Leica Instruments, Bannockburn, IL). Parasagittal slices (300 μM) of the vermis were cut, transferred to 35°C ACSF for 30 min, and then stored at room temperature. The ACSF was identical to the PC cutting solution except that it contained (in mM): 2.5 $CaCl_2$ and 1.3 $MgCl_2$.

## Electrophysiology

Recordings were made from PCs or MLIs visually identified using a 60X water immersion objective on an upright microscope (Scientifica, Uckfield, UK) equipped with a custom-made contrast-imaging gradient (Dodt optics). Electrical responses were measured with a Multiclamp 700A amplifier with pClamp 10 software (Molecular Devices, Sunnyvale, CA), filtered at 2–5 kHz and digitized at 15–50 kHz using a Digidata 1440A AD converter (Molecular Devices). Slices were continuously superfused with ACSF maintained at ~32°C with an inline heater (ALA Scientific Instruments, Farmingdale, NY) at a rate of 2–3 mL/min. ACSF contained 100 μM picrotoxin (Abcam, Cambridge, MA) and other drugs were applied in the bath as indicated.

*Purkinje cell recordings.* Patch pipettes were pulled from thin wall borosilicate glass (Sutter Instruments, Novato, CA) to a resistance of 0.8–2.0 MΩ on a P-97 micropipette puller (Sutter Instruments)

and filled with solution containing (in mM): 110 CsCl, 35 CsF, 10 HEPES, 10 EGTA, and 5 QX-314 adjusted to pH 7.2 with CsOH. Series resistance ($R_s$), measured by the instantaneous current response to a −2 mV step with only pipette capacitance cancelled, was <5 MΩ and was routinely compensated >80%. $R_s$ was monitored throughout the recording and experiments were discarded if substantial changes were observed (>20%). Release mode was manipulated by changing the [$CaCl_2$] and [$MgCl_2$] in the ACSF; for high $Ca^{2+}$ experiments [$Ca^{2+}$] and [$Mg^{2+}$] were 2.5 and 1.3 mM, respectively, and 0.5 and 10 mM, respectively, for most low $Ca^{2+}$ experiments. The [$Mg^{2+}$] was lowered from 10 to 5 mM when the RRP was measured in low $Ca^{2+}$ (*Figure 2D,E*) to enable axons to faithfully fire at 100 Hz. To minimize voltage clamp errors in high $Ca^{2+}$ experiments, cells were held between −10 and −20 mV or at −60 mV in the presence of 3 mM kynurenic acid (Abcam). Cells were held at −60 mV in low $Ca^{2+}$ experiments. CFs were stimulated with theta glass electrodes (BT-150, Sutter Instruments) filled with 5% NaCl driven by a constant current isolated stimulator (Digitimer North America, Ft Lauderdale, FL) and placed in the granule cell layer.

*Molecular layer interneuron recordings.* Patch pipettes were pulled from thick wall borosilicate glass (Sutter Instruments) to a resistance of 2.5–5.0 MΩ. PF minimal stimulation experiments (*Figure 8*) were performed as described in *Nahir and Jahr (2013)*. Cells in the outer 1/3 of the molecular layer were targeted with pipettes filled with a solution containing (in mM): 100 $CsMeSO_3$, 50 CsCl, 10 HEPES, 10 EGTA, 1 $MgCl_2$, 2 MgATP, 0.3 NaGTP, and 5 QX-314, adjusted to pH 7.2 with CsOH. ACSF contained 1.5 mM $Ca^{2+}$ and 1.0 mM $Mg^{2+}$. PFs were isolated using two monopolar stimulators filled with ACSF, which were placed in the granule cell layer within ~20 µM of each other. Their positions and stimulus intensity were adjusted until an all-or-none response was elicited. (R)-CPP (5 µM, Abcam) was included in ACSF for all MLI recordings and cells were held between −60 and −70 mV. Series resistance ($R_s$), measured by the instantaneous current response to a −5 mV step with only pipette capacitance cancelled was 15–20 MΩ. $R_s$ was uncompensated and was monitored throughout the recording. Experiments were discarded if substantial changes were observed (>20%).

## Drug treatments

Slices were incubated with PKA inhibitors (KT5720, 1 µM and 8-Br-cAMPs, 50 µM) for 60–120 min prior to recording and interleaved with untreated control slices. PKA activators (fsk, 50 µM and 6-Bnz, 20 µM) were bath applied during the course of the experiment. To minimize variability in LAA experiments, a single batch of KYN, made and aliquoted at the same time, and a single serial dilution of NBQX were used for each data set. When an increase in % inhibition by the LAA was expected, we selected an LAA concentration that inhibited ~50% of the response in the control treatment group. In contrast, when we predicted a decrease in the % inhibition by the LAA, we used an LAA concentration that inhibited ~70% of the response in control conditions.

## Numerical simulations

We calculated the number of vesicles released ($V_{released}$) using a modified version of the FD (facilitation and depression; FD2 model) model in *Dittman et al. (2000)*. Briefly, we added an additional state transition to account for the effects of PKA-dependent synapsin phosphorylation. Assuming an unlimited number of vesicles and a fixed number of docking sites, the readily releasable pool (RRP)

$$RRP = docking\ sites^* P_{occ}$$

where $P_{occ}$ = probability of site occupancy

$$V_{released} = RRP^* P_{succ}.$$

Simulations with FD2 model were carried out with Mathematica (v11, Wolfram Research, Champaign, IL) using parameters and rates in *Table 1*. The probability of MVR as a function of the number of release sites (*Figure 7Biv*) was calculated from a fraction of a plane horizontal to the hyperbolic paraboloid (Pringles chip) function when $V_{release} > 1, >2, >3$, and >4.

The area $A$ of the part of the plane corresponding to condition $vxy>z$ is equal to the integral

**Table 1.** Parameters used in FD2 model.

| Symbol | Definition | |
|---|---|---|
| $CaX_{F0}$ | Concentration of $Ca^{2+}$-bound site $X_F$ | 0 |
| $CaX_{D0}$ | Concentration of $Ca^{2+}$-bound site $X_D$ | 0 |
| $\Delta F$ | Incremental increase in $CaX_F$ after a stimulus | 5 |
| $\Delta D$ | Incremental increase in $CaX_D$ after a stimulus | 0.001 |
| $\tau_F$ | Decay time constant of $CaX_F$ after an action potential | 0.1 $sec^{-1}$ |
| $\tau_D$ | Decay time constant of $CaX_D$ after an action potential | 0.05 s |
| $K_F$ | Affinity of $CaX_F$ for site | 2 |
| $K_D$ | Affinity of $CaX_D$ for site | 2 |
| $k_0$ | Baseline rate of recovery from recovery state | 0.7 $sec^{-1}$ |
| $k_{max}$ | Maximal rate of recovery from refractory state | 20 $sec^{-1}$ |
| D | Fraction of sites that are release-ready | 1 |
| F | Facilitation probability | (0–1) |
| $N_T$ | Total number of sites | (1 - 10) |
| $P_{occ}$ | Probability of site occupancy | (0–1) |
| RRP | Readily releasable pool | $N_T * P_{occ}$ |
| $P_{succ}$ | Probability that a competent vesicle will release | F * D |

DOI: https://doi.org/10.7554/eLife.47434.018

$$A = \int_{\frac{Z}{v}}^{1} dx \int_{\frac{Z}{vx}}^{1} dy = 1 + \frac{Z}{v}\left(log\frac{Z}{v} - 1\right)$$

The full area $A_0$ of the plane is equal to the integral

$$A_0 = \int_{0}^{1} dx \int_{0}^{1} dy = 1$$

Then the fraction is equal to

$$fraction = \frac{A}{A_0} = 1 + \frac{Z}{v}\left(log\frac{Z}{v} - 1\right)$$

EPSC waveforms (**Figure 7C**) were simulated using the function

$$EPSC = t\frac{e}{\tau E}\left(e^{-\frac{t}{\tau}}\right)$$

where τE is the weighted exponential time constant describing the EPSC (1.5 ms) with the amplitude scaled to represent units of vesicles ($v$) released.

## Quantification and statistical analysis

Data were analyzed using AxoGraph X (Axograph Scientific, Sydney, Australia) and GraphPad Prism (GraphPad, La Jolla, CA). Reported values are mean ± SEM and the statistical test used for each data set is listed in the figure legends. Means were compared using paired or unpaired two-tailed t-tests or 1- or 2-way ANOVAs with Holm-Sidak multiple comparison tests. Cumulative probability histograms were compared using Komogorov-Smirnov tests. Extra sum-of-squares F tests were used to compare curves fit to different groups (drug treatments or genotypes) within the same experiment. The criteria for statistical significance was p<0.05.

## Data and software availability

Mathematica files for the FD2 model have been deposited in ModelDB (senselab.med.yale.edu/modeldb/).

## Acknowledgements

We thank Lynn Dobrunz, Craig Jahr, Anastassios Tzingounis, members of the Wadiche labs for helpful comments throughout this project, and Mary Seelig for technical assistance. This work was supported by NIH NS064025, NS105438 (LOW) and NIH NS065920 (JIW).

## Additional information

### Funding

| Funder | Grant reference number | Author |
| --- | --- | --- |
| National Institute of Neurological Disorders and Stroke | NS065920 | Jacques I Wadiche |
| National Institute of Neurological Disorders and Stroke | NS064025 | Linda Overstreet-Wadiche |
| National Institute of Neurological Disorders and Stroke | NS105438 | Linda Overstreet-Wadiche |

The funders had no role in study design, data collection and interpretation, or the decision to submit the work for publication.

### Author contributions

Jada H Vaden, Conceptualization, Formal analysis, Supervision, Validation, Investigation, Methodology, Writing—original draft, Project administration, Writing—review and editing; Gokulakrishna Banumurthy, Formal analysis, Validation, Investigation, Methodology; Eugeny S Gusarevich, Software, Investigation, Methodology, Writing—review and editing; Linda Overstreet-Wadiche, Conceptualization, Formal analysis, Funding acquisition, Investigation, Methodology, Project administration; Jacques I Wadiche, Conceptualization, Resources, Formal analysis, Supervision, Funding acquisition, Validation, Investigation, Methodology, Writing—original draft, Project administration, Writing—review and editing

### Author ORCIDs

Jada H Vaden (iD) https://orcid.org/0000-0002-1295-0236
Eugeny S Gusarevich (iD) https://orcid.org/0000-0002-8642-6293
Linda Overstreet-Wadiche (iD) https://orcid.org/0000-0001-7367-5998
Jacques I Wadiche (iD) https://orcid.org/0000-0001-8180-2061

### Ethics

Animal experimentation: All experiments were conducted through protocols approved by the Institutional Animal Care and Use Committee of the University of Alabama at Birmingham under protocol #08767.

### Decision letter and Author response

Decision letter https://doi.org/10.7554/eLife.47434.023
Author response https://doi.org/10.7554/eLife.47434.024

## Additional files

### Supplementary files

• Transparent reporting form

DOI: https://doi.org/10.7554/eLife.47434.019

## Data availability

All data generated or analyzed during this study are included in the manuscript and supporting files. Mathematica files for the FD2 model have been deposited in ModelDB (https://senselab.med.yale.edu/modeldb/).

The following dataset was generated:

| Author(s) | Year | Dataset title | Dataset URL | Database and Identifier |
|---|---|---|---|---|
| Vaden, Banumurthy, Gusarevich, Overstreet-Wadiche, Wadiche | 2019 | Vesicle release model | https://senselab.med.yale.edu/modeldb/258843 | ModelDB, 258843 |

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
