## [Decision Letter]

Thank you for submitting your article "The readily-releasable pool dynamically regulates multivesicular release" for consideration by *eLife*. Your article has been reviewed by three peer reviewers, and the evaluation has been overseen by a Reviewing Editor and Richard Aldrich as the Senior Editor. The reviewers have opted to remain anonymous.

The reviewers have discussed the reviews with one another and the Reviewing Editor has drafted this decision to help you prepare a revised submission.

Summary:

In this study, the authors present compelling evidence for the divergence between regulation of vesicle release probability and the readily releasable pool (RRP) size in regulation of multivesicular release. In well controlled experiments it is shown that activation of protein kinase A (PKA) increases multivesicular release via an increase in the readily releasable pool but not via an increase in release probability (Pr) at climbing fiber synapses in low external calcium, a condition that promotes UVR and low Pr. PKA activity can thus change RRP without altering Pr. The authors then link the effects of PKA to synapsin using triple KO mice. They also perform experiments in the parallel fiber synapse of the cerebellum and they develop a mathematical model to explain their results that uses two distinct release probability parameters: a P_succ_ and a P_occ_ of release sites.

Essential revisions:

Generally, the reviewers are very positive about the work and recommend publication provided that the following concerns are addressed. None of the points was considered as absolutely essential for revision, and we therefore include the relevant parts of the reviewer comments in an unedited version.

Reviewer #1:

1) Sr^2+^ experiments: The authors measure inter-event intervals up to 600-700 ms. What was the time window after each stimulation used for these measurements? How clean is the separation between the rate of aEPSCs and mEPSCs?

2) Does the synchronous component of the EPSCs evoked in Sr^2+^ show cAMP-PKA mediated potentiation as well? I think this is important, in order to validate that Sr^2+^ driven release reports the same basic process as Ca^2+^ driven release (besides being more desynchronized).

3) A potential explanation for these observations could be compound fusion (or simply having larger vesicles). Sr^2+^ experiments argue against these possibilities as well.

4) Are there any changes in the rates of RRP depletion and recovery (after cAMP-PKA manipulation or in synapsin TKOs)? Authors may already have this data. I think these parameters may help elucidate what the exact target of synapsin action is.

*Reviewer #2:*

Subsection “cAMP/PKA stimulation shifts the balance of vesicle release from UVR to MVR mode without affecting Pr” and Materials and methods: The authors use P12-18 mice for their studies and these are quite immature and probably to not have fully mature cerebellar circuits and synapses. Mice at P12 are barely able to hear (ear canal opens at this age) and eye opening occurs only at P14. It would thus be important for the authors to do an analysis of the results of Figure 1 in P12-13 mice and P17-18 mice separately and see if there are any differences in the experimental results. Are the synapses at P12 fully mature and identical to P18 so that one can group them together? It would also be interesting to know if more mature P30 climbing fiber synapses also have a similar increase to forskalin and RRP size as P12 and P18 synapses.

Introduction: The authors should provide some numbers for Pr at different synapses. What is high Pr and what is low? This would be helpful to the general reader. At cultured hippocampal synapses the Pr = 0.05 at some synapses and at the climbing fiber synapse Silver et al. (1998) estimate Pr = 0.9. Please add also the paper of Taschenberger et al. (2002) to the list of low Pr (= 0.3) synapses that exhibit MVR (young calyx of Held synapse) together with Oertner et al. (2002).

Subsection “PKA-inhibition shifts vesicle release mode from MVR to UVR”: Is there evidence for PKA expression in the postsynaptic Purkinje cell dendrites? This should be cited from the literature, if its available. The lack of effect on the mEPSC amplitude is a good control on possible effects of PKA on AMPARs.

Subsection “cAMP/PKA inhibition reduces the size of the RRP without Pr changes”: Add reference to Silver et al. (1998) together with Foster and Regehr reference since they are the first to show high Pr and vesicle pool depletion at climbing fiber synapses. Also add reference to Taschenberger et al. (2002) together with Elmquist and Quastel (1965) since they were the first to use this method in CNS synapses to estimate RRP size at rapidly depressing synapses that show vesicle pool depletion.

*Reviewer #3:*

1) Figure 1—figure supplement 2 suggests that synaptic depression during repetitive stimulation involves reduction in the number of active release sites. However, by comparing the stand-alone EPSCs which are blocked by KYN by 60-70% , the steady-state EPSCs after repetitive stimulation are blocked by KYN by > 80% . This suggests that effective glutamate concentration in the cleft is reduced during repetitive stimulation, arguing for reduction in release probability within active zones. The authors should comment on this issue.

2) Use of cumulative release during train stimulation for estimating the RRP size could be a matter of discussion (Neher, 2015). Though the authors' arguments are most likely correct, they should show the time course of the EPSC amplitudes during the train (average data and normalized data). If the time course is unchanged, it is unlikely that Pr is modulated, but the RRP size is changed.

3) The link between the external Ca^2+^ and PKA activation is unclear. Is it possible that incubation of the slice with EGTA-AM occludes the effects of KT5720 under high external condition? Else, is it possible that the effects of EGTA on the evoked EPSCs are not due to the loose coupling between Ca^2+^ channels and synaptic vesicles but rather due to activation of PKA by basal Ca^2+^? Either the authors address this issue experimentally or discuss the possibilities.

4) Although mEPSC amplitudes were unchanged, it would be nice if the authors have additional evidence that KT5720 and cAMP do not change AMPA receptor properties (responsible for the evoked EPSCs), given that PKA is known to modulate postsynaptic receptors. At least the authors discuss possible postsynaptic modulation and exclude the possibility.

---

## [Author Response]

Essential revisions:Generally, the reviewers are very positive about the work and recommend publication provided that the following concerns are addressed. None of the points was considered as absolutely essential for revision, and we therefore include the relevant parts of the reviewer comments in an unedited version.Reviewer #1:1) Sr^2+^ experiments: The authors measure inter-event intervals up to 600-700 ms. What was the time window after each stimulation used for these measurements? How clean is the separation between the rate of aEPSCs and mEPSCs?

We detected asynchronous events for 700 ms following stimulation, although most (~99%) occurred in the first 500 ms. The frequency of events in the 50 ms window before the stimulus was very low (0.32 Hz on average), so we expect that the contribution of spontaneous events to our results is minimal.

2) Does the synchronous component of the EPSCs evoked in Sr^2+^ show cAMP-PKA mediated potentiation as well? I think this is important, in order to validate that Sr^2+^ driven release reports the same basic process as Ca^2+^ driven release (besides being more desynchronized).

There was a small but significant increase in EPSC amplitude in 4 cells (from 6.04 ± 1.7 nA to 6.63 ± 1.8 nA, p = 0.02, paired t-test). The synchronous EPSC saturated the amplifier in the 2 remaining cells from this data set, and could not be measured.

3) A potential explanation for these observations could be compound fusion (or simply having larger vesicles). Sr^2+^ experiments argue against these possibilities as well.

We agree that regulation of compound fusion or the amount of glutamate per vesicle by PKA could explain the effects of blocking or activating PKA on the inhibition of EPSCs by kynurenic acid. However, as the reviewer points out, we would expect to see corresponding effects on aEPSCs. Moreover, the effects of manipulating PKA are lost in synapsin TKOs and we aren’t aware of any evidence linking synapsins to compound fusion or the amplitude of quantal events.

4) Are there any changes in the rates of RRP depletion and recovery (after cAMP-PKA manipulation or in synapsin TKOs)? Authors may already have this data. I think these parameters may help elucidate what the exact target of synapsin action is.

We found no difference in the rates of depletion during train stimulation when we activated PKA. We have now included a supplementary figure (Figure 2—figure supplement 1) showing the time course of EPSC amplitudes during train stimulation before and 6Bnz-cAMP. No differences were found in the facilitating and depressing time course of EPSC amplitudes.

Similarly, Figure 5B (lower inset) shows the rate of depletion did not change following PKA inhibition (KT5720). The time course of depletion was best fit with a single exponential curve that did not differ between control and KT5720-treated slices. We have now included these numbers in the legend for Figure 5.

Since Pr at the CF is very high in 2.5 mM Ca^2+^, a single stimulus depletes much of the RRP. We therefore estimated recovery from depletion by assaying paired pulse depression across a range of interstimulus intervals. These data are shown in Figure 4—figure supplement 1, along with the two-phase exponential decay curves fit to the averaged data. Individual fits did not differ between control and KT-treated slices. We have now included these numbers in the legend for Figure 4—figure supplement 1.

Reviewer #2:Subsection “cAMP/PKA stimulation shifts the balance of vesicle release from UVR to MVR mode without affecting Pr” and Materials and methods: The authors use P12-18 mice for their studies and these are quite immature and probably to not have fully mature cerebellar circuits and synapses. Mice at P12 are barely able to hear (ear canal opens at this age) and eye opening occurs only at P14. It would thus be important for the authors to do an analysis of the results of Figure 1 in P12-13 mice and P17-18 mice separately and see if there are any differences in the experimental results. Are the synapses at P12 fully mature and identical to P18 so that one can group them together? It would also be interesting to know if more mature P30 climbing fiber synapses also have a similar increase to forskalin and RRP size as P12 and P18 synapses.

We performed our experiments in young mice to improve tissue health and voltage clamp. CFs begin dendritic translocation at P12 and are in the late phase of functional differentiation from P12-P17 (Watanabe and Kano, 2011). We only recorded from cells in which a single CF could be reliably isolated. We now note this in the Materials and methods subsection “Experimental Model and Subject Details”.

We also recorded CF-mediated spillover onto molecular layer interneurons (MLIs) in mice aged P21-28 and found that KT5720 and Rp-8-Br-cAMPs increased sensitivity of the spillover EPSCs to kynurenic acid without altering short-term plasticity (Author response image 1). This result shows that manipulating PKA in older CFs is qualitatively similar to what we observed in younger animals (Figure 4A-C). We have not included this figure in our final manuscript because it was performed in MLIs rather than PCs.

**Author response image 1. respfig1:** PKA inhibition reduces the glutamate transient underlying spillover response onto MLIs in mice aged P21-P28. (**A**) Superimposed CF-MLI spillover EPSCs before and after bath application of γ-DGG (300 µM) or NBQX (200 nM) in control slices and after incubation with KT5720 (1 µM) or 8-Br-cAMPs (50 µM) for 90-120 min. (**B**) KT5720 and 8-BrcAMPs increased sensitivity to γDGG (control: 36.1% ± 5.5, n = 9; KT5720: 72.1% ± 4.2, n = 5, p = 0.0006; Rp-8-Br-cAMPs: 63.6% ± 4.3, n = 5, p = 0.0044), but not NBQX (control: 50.4 ± 2.8, n = 4; KT5720: 53.2 ± 8.4, n = 3, p = 0.71; Rp-8-Br-cAMPs: 57.78% ± 2.5, n = 5, p = 0.49). One-way ANOVA and Holm-Sidak post-tests. (C, left) Normalized representative EPSCs in response to paired stimuli (100 ms) in control-, KT5720-, and 8-Br-cAMPs-treated slices. (C, right)Summary of PPR in control- (0.21 ± 0.02, n = 12), KT5720- (0.23 ± 0.03, n = 7, p > 0.99) and 8-BrcAMPs-treated slices (0.26 ± 0.03, n = 7, p >0.99). One-way ANOVA and Holm-Sidak post-tests.

Introduction: The authors should provide some numbers for Pr at different synapses. What is high Pr and what is low? This would be helpful to the general reader. At cultured hippocampal synapses the Pr = 0.05 at some synapses and at the climbing fiber synapse Silver et al. (1998) estimate Pr = 0.9. Please add also the paper of Taschenberger et al. (2002) to the list of low Pr (= 0.3) synapses that exhibit MVR (young calyx of Held synapse) together with Oertner et al. (2002).

We have added the requested reference and included a range of Prs in the second paragraph of the Introduction, which we agree is helpful to readers.

Subsection “PKA-inhibition shifts vesicle release mode from MVR to UVR”: Is there evidence for PKA expression in the postsynaptic Purkinje cell dendrites? This should be cited from the literature, if its available. The lack of effect on the mEPSC amplitude is a good control on possible effects of PKA on AMPARs.

We have added a paragraph in the Discussion that includes references related to postsynaptic PKA effects as well as our extensive evidence against modulation of AMPARs mediating our effects.

Subsection “cAMP/PKA inhibition reduces the size of the RRP without Pr changes”: Add reference to Silver et al. (1998) together with Foster and Regehr reference since they are the first to show high Pr and vesicle pool depletion at climbing fiber synapses. Also add reference to Taschenberger et al. (2002) together with Elmquist and Quastel (1965) since they were the first to use this method in CNS synapses to estimate RRP size at rapidly depressing synapses that show vesicle pool depletion.

We have added these references.

Reviewer #3:1) Figure 1—figure supplement 2 suggests that synaptic depression during repetitive stimulation involves reduction in the number of active release sites. However, by comparing the stand-alone EPSCs which are blocked by KYN by 60-70% , the steady-state EPSCs after repetitive stimulation are blocked by KYN by > 80% . This suggests that effective glutamate concentration in the cleft is reduced during repetitive stimulation, arguing for reduction in release probability within active zones. The authors should comment on this issue.

The concentrations of KYN and external Mg^2+^ differed between the two experiments, so the inhibition by KYN cannot be directly compared. In Figure 1, we used 0.5 mM Ca^2+^ and 10 mM Mg^2+^ to constrain release to UVR. KYN (250 µM) was used in these experiments because it blocks ~60% of the response in control conditions, allowing us to detect either an increase or decrease in sensitivity to KYN.

In contrast, in Figure 1—figure supplement 2, we tested whether changes in the density of release sites altered inhibition by KYN. We were concerned that neurotransmitter pooling between synapses, that is more likely to occur if multiple vesicles are released, could affect our interpretation. We therefore decreased the Mg^2+^ concentration to 3.3 mM to allow some MVR in our control conditions (0.1 Hz stimulation). The concentration of KYN used in this experiment was 1 mM (as in Rudolph et al., 2015).

2) Use of cumulative release during train stimulation for estimating the RRP size could be a matter of discussion (Neher, 2015). Though the authors' arguments are most likely correct, they should show the time course of the EPSC amplitudes during the train (average data and normalized data). If the time course is unchanged, it is unlikely that Pr is modulated, but the RRP size is changed.

There was no difference in the rates of depletion during train stimulation when we activated or inhibited PKA. See reviewer 1, point 4.

3) The link between the external Ca^2+^ and PKA activation is unclear. Is it possible that incubation of the slice with EGTA-AM occludes the effects of KT5720 under high external condition? Else, is it possible that the effects of EGTA on the evoked EPSCs are not due to the loose coupling between Ca^2+^ channels and synaptic vesicles but rather due to activation of PKA by basal Ca^2+^? Either the authors address this issue experimentally or discuss the possibilities.

One possibility is that external Ca^2+^ affects AC/PKA activity at rest in a manner independent of Ca^2+^ influx. At least for cultured cells, it isn’t clear that exposure to high external Ca^2+^ causes a corresponding increase in intracellular Ca^2+^ at rest (Vyleta and Smith, 2011), so pre-incubation with EGTA-AM may not affect PKA activation. The same study shows that external Ca^2+^ interacts with membrane bound, Ca^2+^ sensitive receptors (CaSRs) that translate changes in external Ca^2+^ to internal signaling. In this case, we expect that the effects of raising external Ca^2+^ will be independent of Ca^2+^ chelation. We now have now included this possibility in our Discussion (subsection “Synapsins, RRP, and synaptic transmission”) and think that this will be interesting to test in the future.

4) Although mEPSC amplitudes were unchanged, it would be nice if the authors have additional evidence that KT5720 and cAMP do not change AMPA receptor properties (responsible for the evoked EPSCs), given that PKA is known to modulate postsynaptic receptors. At least the authors discuss possible postsynaptic modulation and exclude the possibility.

We have now included a section on the possibility of postsynaptic modulation in the Discussion.